# Systematic analysis of the IL-17 receptor signalosome reveals a robust regulatory feedback loop

Helena Draberova[1,2,†], Sarka Janusova[2,†], Daniela Knizkova[1,2,†], Tereza Semberova[1,2], Michaela Pribikova[1,2], Andrea Ujevic[1,2], Karel Harant[3], Sofija Knapkova[4,5], Matous Hrdinka[4,5] [ID], Viola Fanfani[6], Giovanni Stracquadanio[6], Ales Drobek[2] [ID], Klara Ruppova[2], Ondrej Stepanek[2,*] [ID] & Peter Draber[1,2,**] [ID]

## Abstract

**IL-17 mediates immune protection from fungi and bacteria, as well as it promotes autoimmune pathologies. However, the regulation of the signal transduction from the IL-17 receptor (IL-17R) remained elusive. We developed a novel mass spectrometry-based approach to identify components of the IL-17R complex followed by analysis of their roles using reverse genetics. Besides the identification of linear ubiquitin chain assembly complex (LUBAC) as an important signal transducing component of IL-17R, we established that IL-17 signaling is regulated by a robust negative feedback loop mediated by TBK1 and IKKε. These kinases terminate IL-17 signaling by phosphorylating the adaptor ACT1 leading to the release of the essential ubiquitin ligase TRAF6 from the complex. NEMO recruits both kinases to the IL-17R complex, documenting that NEMO has an unprecedented negative function in IL-17 signaling, distinct from its role in NF-κB activation. Our study provides a comprehensive view of the molecular events of the IL-17 signal transduction and its regulation.**

**Keywords** IKKε; IL-17; LUBAC; NEMO; TBK1
**Subject Categories** Immunology; Signal Transduction
**The EMBO Journal (2020) 39: e104202**

## Introduction

The interleukin 17 (IL-17) is a major proinflammatory cytokine produced by Th17 cell lineage and several innate immune cell types (Harrington *et al*, 2005; Park *et al*, 2005; Cua & Tato, 2010). Studies of mouse models demonstrated that this cytokine is crucial for host defense against opportunistic fungal and bacterial species (Conti *et al*, 2009; Cho *et al*, 2010). In accord, patients impaired in IL-17 signaling suffer from chronic mucocutaneous candidiasis (Puel *et al*, 2011; Conti & Gaffen, 2015). On the other hand, aberrant signaling via IL-17 promotes pathogenesis of several autoimmune disorders, such as psoriasis, atopic dermatitis, rheumatoid arthritis, or multiple sclerosis (Brembilla *et al*, 2018) and therapeutic antibodies blocking IL-17 or its receptor have been successfully used in clinic to treat severe plaque psoriasis (Bilal *et al*, 2018; Hawkes *et al*, 2018). Altogether, IL-17 production and signal transduction must be subjected to a tight control to allow proper immune system response when required, yet preventing autoinflammatory diseases.

Interleukin-17 receptor (IL-17R) is composed of two widely expressed subunits IL-17RA and IL-17RC (Toy *et al*, 2006; Hu *et al*, 2010). Binding of dimeric IL-17 leads to heterodimerization of the receptor (Ely *et al*, 2009; Liu *et al*, 2013; Goepfert *et al*, 2017) and recruitment of a cytoplasmic protein ACT1 (Chang *et al*, 2006; Qian *et al*, 2007). ACT1 was described to enhance the expression of genes encoding proinflammatory cytokines by stabilizing their mRNAs or by activating downstream signaling pathways leading to the activation of their transcription (Li *et al*, 2019). The gene-activation pathways are dependent on the recruitment of E3 ubiquitin ligases from tumor necrosis factor (TNF) receptor-associated factors (TRAFs) family, most prominently TRAF6. TRAF6 creates non-degradative K63-polyubiquitin linkages which serve as docking sites for a variety of signaling molecules and promote activation of downstream signaling pathways, especially mitogen-activated protein kinase (MAPK) or nuclear factor-κB (NF-κB) and subsequent production of

1 Laboratory of Immunity & Cell Communication, BIOCEV, First Faculty of Medicine, Charles University, Vestec, Czech Republic
2 Laboratory of Adaptive Immunity, Institute of Molecular Genetics of the Czech Academy of Sciences, Prague, Czech Republic
3 Laboratory of Mass Spectrometry, BIOCEV, Faculty of Science, Charles University, Prague, Czech Republic
4 Department of Haematooncology, University Hospital Ostrava, Ostrava, Czech Republic
5 Faculty of Medicine, University of Ostrava, Ostrava,Czech Republic
6 Institute of Quantitative Biology, Biochemistry, and Biotechnology, SynthSys, School of Biological Sciences, University of Edinburgh, Edinburgh, UK
  *Corresponding author. Tel: +42 0241 062155; E-mail: ondrej.stepanek@img.cas.cz
  **Corresponding author. Tel: +42 0735 208125; E-mail: peter.draber@lf1.cuni.cz
  †These authors contributed equally to this work

proinflammatory cytokines (Schwandner *et al*, 2000; Sonder *et al*, 2011). However, the mechanisms promoting and regulating IL-17 signaling emanating directly from IL-17 receptor are incompletely understood. The IL-17-induced activation of downstream pathways is surprisingly weak in comparison with other proinflammatory cytokines such as IL-1α or TNF, although their receptors all employ the formation of non-degradative polyubiquitin linkages and share multiple proximal signaling proteins (Kupka *et al*, 2016; Strickson *et al*, 2017; Li *et al*, 2019; McGeachy *et al*, 2019). The molecular basis for these differences is poorly defined. In addition to directly inducing activation of signaling pathways, IL-17 can trigger stabilization of mRNA transcripts via ACT1 and TRAF2/5, which regulate mRNA stability either directly, or by modulating the activity of mRNA binding proteins ARID5A and HuR, splicing factor SF2, and endoribonuclease Regnase-1 (Sun *et al*, 2011; Herjan *et al*, 2013, 2018; Somma *et al*, 2015; Amatya *et al*, 2018).

In this study, we established a novel methodical approach to analyze the assembly of the IL-17 receptor signaling complex (IL-17RSC) via mass spectrometry (MS), which revealed the composition of the complex and its stoichiometry, including a novel signaling mediator, linear ubiquitin chain assembly complex (LUBAC). Importantly, we uncovered a robust negative inhibitory loop mediated by NEMO-recruited kinases TBK1 and IKKε that is specific for the IL-17 pathway, explaining the enigmatic mechanism of a weak signaling response of cells to IL-17 stimulation and showing a unique regulatory role of NEMO in the assembly of IL-17RSC.

# Results

## Kinases TBK1 and IKKε are strongly and preferentially activated upon IL-17 stimulation

We aimed to resolve the composition of the IL-17 receptor signaling complex formed upon the binding of IL-17 to its receptors. For that purpose, we deployed a strategy for receptor-complex analysis in which cells were stimulated with a recombinant dimeric IL-17 (Fig EV1A–C), followed by the pull-down of the whole signaling complex via the ligand's tandem affinity purification tag (2xStrep-tag and 1xFlag-tag) and MS analysis (Fig 1A). This approach offers the possibility to isolate only ligand-engaged receptors forming membrane-proximal signaling complexes via highly specific tandem affinity purification without the requirement for exogenous expression of tagged proteins in target cells. As a control, the ligand was added after the cell lysis, which did not induce assembly of the signaling complex. The IL-17 stimulation might lead to post-translational modifications of potential contaminants that would change their binding to the beads used for immunoprecipitation. In order to ensure that the identified proteins are bona fide components of the IL-17RSC, we decided for relatively strict definition of background contaminants (as described in Table EV1).

The comparison between control and stimulated samples revealed a very specific and highly reproducible set of proteins recruited to IL-17RSC. In contrast to IL-17RA, we detected IL-17RC only in stimulated but not control samples, which reflects that murine IL-17RC binds IL-17 only when it is associated with IL-17RA (Kuestner *et al*, 2007). Importantly, we identified a number of previously known components of IL-17RSC: core protein ACT1, non-

degradative ubiquitin ligases TRAF6 and TRAF2, deubiquitinase A20 and associated adaptors ABIN1 and TAX1BP1, a kinase complex NEMO/IKKα/IKKβ, and homologous kinases TBK1 and IKKε (Amatya *et al*, 2017). We also identified components of a degradative ubiquitin ligase complex consisting of βTrCP1/2 and Cullin1, which were previously reported to degrade ACT1 upon prolonged stimulation (Shi *et al*, 2011), although it was not known they are recruited directly to the IL-17RSC. In addition, we identified proteins TANK and NAP1 that have not yet been connected to the IL-17R pathway (Fig 1B and Table EV1). These two adaptors were reported to associate with TBK1 and IKKε (Chau *et al*, 2008; Helgason *et al*, 2013) and recruit them to the TNFR1 signaling complex (TNF-RSC) (Lafont *et al*, 2018).

We subsequently calculated the stoichiometry between individual components of the complex using intensity-based absolute quantification (iBAQ) (Schwanhausser *et al*, 2011). Murine IL-17 binds first strongly to IL-17RA and only subsequently can interact with IL-17RC to form the complex in 1:2:1 stoichiometry between IL-17RA: IL-17:IL-17RC (Ely *et al*, 2009; Liu *et al*, 2013; Goepfert *et al*, 2017). As IL-17RC does not bind directly to IL-17 in the post-lysis control samples, we normalized the iBAQ values of individual proteins to IL-17RC (Fig 1B). Surprisingly, TBK1 and IKKε were among the most abundant components of the complex, largely exceeding the related kinases IKKα and IKKβ that are crucial for NF-κB activation (Fig 1C and Table EV1). We confirmed that both TBK1 and IKKε were recruited and phosphorylated on their activation Ser172 residue (Kishore *et al*, 2002; Ma *et al*, 2012) within the IL-17RSC (Fig EV1D). High abundance of TBK1 and IKKε in the IL-17RSC suggested that their activation might be a major signaling event triggered by IL-17 stimulation. Indeed, IL-17 strongly activated TBK1 and IKKε at a comparable or even higher level as the stimulation with strong proinflammatory stimuli TNF or IL-1α (Fig 1D). In a sharp contrast, NF-κB and MAPK signaling pathways were only weakly triggered by IL-17. The same results were obtained in human cell line HeLa (Fig EV1E). Altogether, these data established that IL-17 shows a unique preference for strong activation of TBK1 and IKKε kinases over other signaling events.

## Kinases TBK1 and IKKε negatively regulate IL-17 signaling

Although our data showed that the activation of TBK1 and IKKε is likely the most prominent signaling event upon IL-17 stimulation, the role of TBK1 and IKKε kinases in IL-17 signaling is highly controversial. Ablation of IKKε or TBK1 alone was described to weakly inhibit MAPK signaling and IL-17-mediated stabilization of mRNA (Bulek *et al*, 2011; Herjan *et al*, 2018). In accord, stimulation of TBK1 and IKKε DKO cells with IL-17 in the presence of TNF led to markedly decreased transcriptional response, indicating that both kinases are positive regulators of IL-17 signaling responses (Tanaka *et al*, 2019). In striking contrast, overexpression of either TBK1 or IKKε in cells deficient for both these kinases led to inhibition of IL-17-induced downstream signaling (Qu *et al*, 2012), indicating that they are in fact negative regulators of IL-17 signaling.

In order to resolve the controversial issue concerning the role of these kinases in shaping IL-17 responses, we employed RNA sequencing to analyze the transcription response following IL-17 stimulation in the presence or absence of MRT67307, a highly specific inhibitor of both TBK1 and IKKε (Clark *et al*, 2011a) (Table EV2

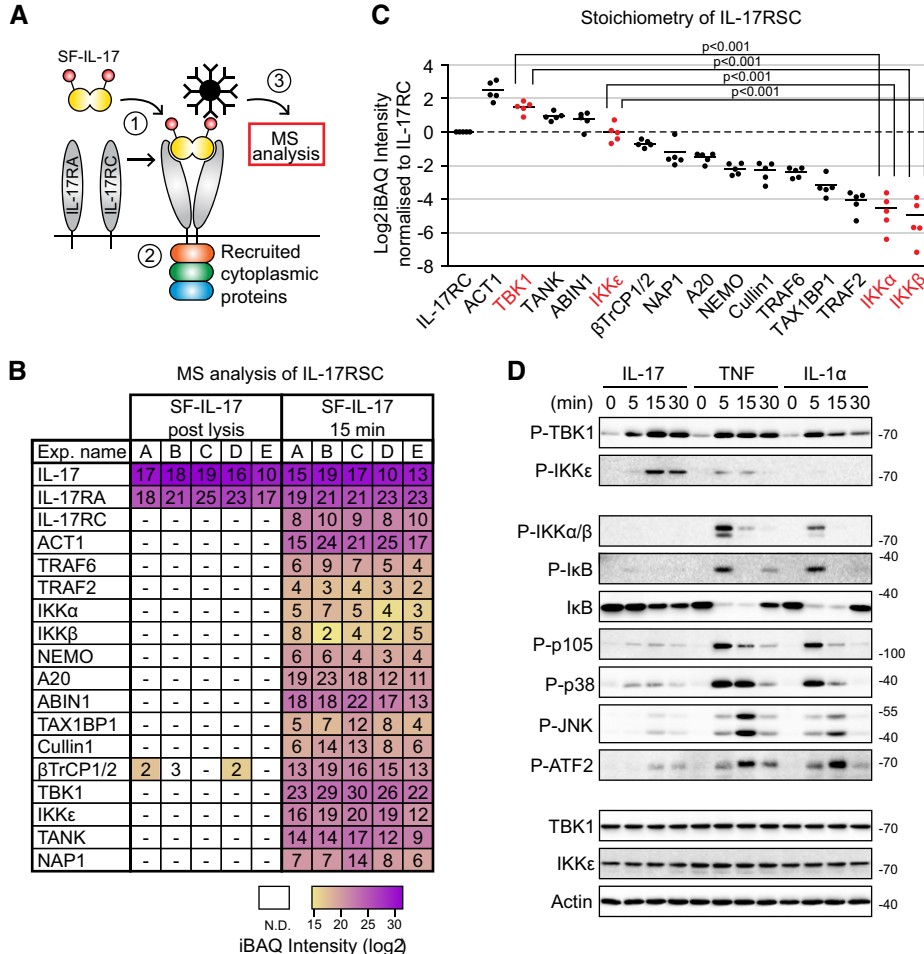

**Figure 1. Kinases TBK1 and IKKε are major components of IL-17RSC.**

A   Schematic representation of IL-17RSC isolation and analysis. Cells are stimulated with recombinant Strep-Flag-IL-17 (SF-IL-17) (1), which leads to the crosslinking of its two receptor subunits and recruitment of cytoplasmic molecules (2). The whole complex is isolated upon cell lysis via tandem affinity purification of the ligand and analyzed by MS (3).

B   ST2 cells were stimulated for 15 min with SF-IL-17 (500 ng/ml), solubilized and IL-17RSC was isolated via consecutive Flag and Strep immunoprecipitation and analyzed by MS. As a control, cells were first solubilized and SF-IL-17 was added only post-lysis. Number of identified peptides (unique + razor) and iBAQ intensities for each protein in five independent experiments are shown.

C   The stoichiometry of IL-17RSC calculated as the ratio between iBAQ intensities of individual IL-17RSC components to iBAQ intensity of IL-17RC. The recruitment of kinases TBK1 and IKKε as compared to related kinases IKKα and IKKβ is significantly enhanced. Mean from five performed MS experiments is shown, the statistical significance was determined using unpaired two-tailed Student's *t*-test.

D   ST2 cells were stimulated with IL-17 (500 ng/ml), TNF (50 ng/ml), or IL-1α (50 ng/ml) for indicated time points and activation of signaling pathways was analyzed by immunoblotting. A representative of two independent experiments is shown.

Source data are available online for this figure.

and Fig EV2A). The comparison of unstimulated with IL-17 stimulated cells showed upregulation of 65 genes, most of them being established targets of the IL-17 signaling pathway (Fig 2A). Treatment of cells with MRT67307 alone induced subtle alternations of the transcriptome that were largely non-overlapping with the effects of IL-17 treatment (Fig EV2B and C). However, IL-17-induced pronounced changes in the transcriptional response of cells pretreated with MRT67307 (Fig 2B). Specifically, the inhibition of TBK1 and IKKε augmented the upregulation of almost all IL-17 responsive genes (Fig 2C ans D). Moreover, multiple IL-17 responsive genes (such as Tnf or Cxcl2) reached the significant level of upregulation only when the IL-17 stimulation was performed in the

presence of MRT67307 (Fig 2E). Real-time PCR analysis confirmed that inhibition of TBK1 and IKKε markedly enhanced the IL-17-mediated upregulation of selected target genes after 2, 4, and 8 h of stimulation (Fig 2F and EV2D). Altogether, these data demonstrated that TBK1 and IKKε kinase activities lead to the general inhibition of IL-17 transcriptional responses.

In the next step, we probed the role of TBK1 and IKKε in the IL-17-triggered proximal signaling pathways. The inhibition of TBK1 and IKKε activity dramatically enhanced the activation of NF-κB and MAPKs (Fig 2G). Subsequently, we prepared cells deficient in TBK1, IKKε, or both using CRISPR/Cas9 approach. Ablation of either kinase alone led to weak suppression of responses to IL-17

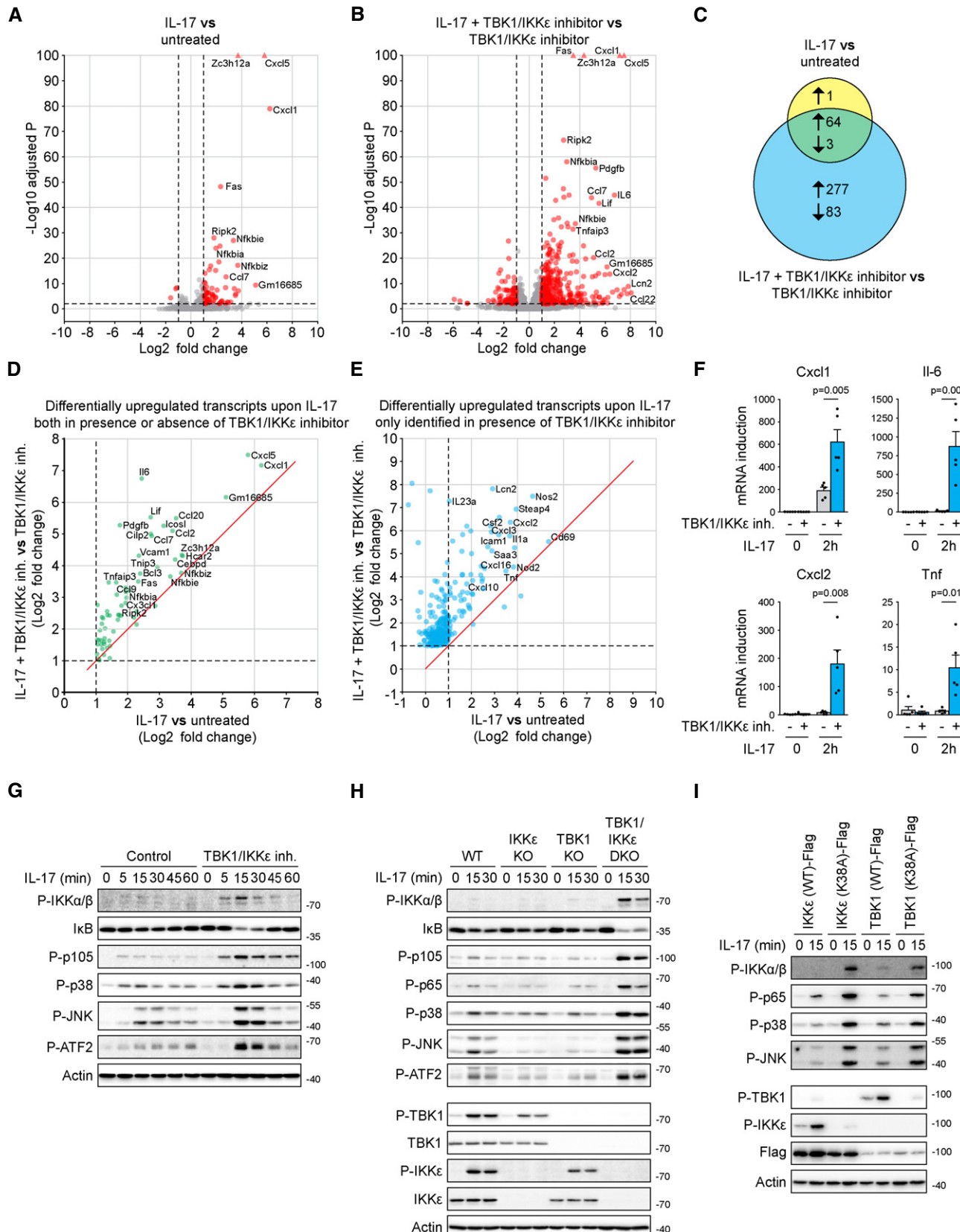

Figure 2.

**Figure 2. TBK1 and IKKε function in redundant manner as inhibitors of IL-17-induced signaling responses.**

A, B   ST2 cells were either left untreated (A) or pretreated with TBK1/IKKε inhibitor MRT67307 (2 μM) for 30 min (B), followed by stimulation with IL-17 (500 ng/ml) for
       2 h. Transcription response induced by IL-17 stimulation as compared to unstimulated cells was analyzed by RNA sequencing. In red are transcripts considered to
       be significantly changed (log₂ fold change > 1 or < −1, −log₁₀ Benjamini–Hochberg adjusted *P*-value > 2, based on the analysis of three independent experiments;
       tringle is used for transcripts with −log₁₀ adjusted *P*-value > 100). Names of several significantly upregulated transcripts are indicated.
C      The Venn diagram representing the number of significantly changed transcripts upon IL-17 stimulation detected in (A) and (B).
D, E   Comparison of IL-17-induced transcriptional response in the presence or absence of TBK1/IKKε inhibitor. (D) Transcripts that are significantly changed in both
       conditions. (E) Transcripts that pass significance threshold for induction only in the presence of TBK1/IKKε inhibitor. Dashed lines indicate significantly upregulated
       transcripts (log₂ fold change > 1); red lines separate transcripts that are more upregulated upon IL-17 stimulation in the presence of TBK1/IKKε inhibitor as
       compared to IL-17 alone.
F      ST2 cells pretreated or not with TBK1/IKKε inhibitor MRT67307 (2 μM) were left unstimulated or stimulated with IL-17 (500 ng/ml) for 2 h and induction of mRNA
       for selected genes was analyzed by real-time PCR. Mean + SEM from five independent experiments is shown, and statistical significance was determined using
       unpaired two-tailed Student's *t*-test.
G      ST2 cells were pretreated or not with TBK1/IKKε inhibitor MRT67307 (2 μM), stimulated with IL17 (500 ng/ml) for indicated time points and analyzed by
       immunoblotting.
H      ST2 wild-type, TBK1 KO, IKKε KO, or cells lacking both kinases (DKO) were stimulated with IL-17 (500 ng/ml) as indicated and analyzed by immunoblotting.
I      TBK1/IKKε DKO cells reconstituted with either wild-type or kinase dead mutant (K38A) version of both kinases were stimulated with IL-17 (500 ng/ml) as indicated
       and analyzed by immunoblotting.

Data information: Immunoblot results are representative of four (G) or three (H, I) independent experiments.
Source data are available online for this figure.

stimulation. In contrast, deficiency in both kinases led to strikingly enhanced activation of major signaling pathways, demonstrating absolute functional redundancy between TBK1 and IKKε in the inhibition of proximal IL-17 signaling (Fig 2H). The reconstitution of TBK1/IKKε double knockout (DKO) cells with either wild-type kinases TBK1 or IKKε, but not with their catalytically inactive versions, led to a strong inhibition of signaling (Fig 2I). As previously reported (Lafont *et al*, 2018), the ablation of both TBK1 and IKKε did not enhance signaling upon TNF, demonstrating that these two kinases have a unique role in the negative regulation of the IL-17 signaling pathway (Fig EV2E). We observed strong TBK1/IKKε activation over a broad range of IL-17 concentrations, which correlated with enhanced signaling in TBK1/IKKε DKO cells (Fig EV2F). Finally, we confirmed that concomitant deletion of TBK1 and IKKε or their chemical inhibition led to drastically enhanced IL-17 signaling also in human HeLa cells (Fig EV2G and H). Altogether, our experimental evidence demonstrates that TBK1 and IKKε are strongly activated upon IL-17 stimulation to provide potent inhibition of the downstream signaling in completely redundant manner.

**TBK1 and IKKε inhibit the recruitment of effector molecules into the IL-17RSC**

To elucidate the molecular mechanism of how TBK1 and IKKε inhibit the IL-17 signaling pathway, we compared the composition of IL-17RSC in WT and TBK1/IKKε DKO cells via MS (Table EV3). The principal component analysis demonstrated that the absence of these two kinases led to markedly changed IL-17RSC composition (Fig EV3A). We observed strong enrichment of non-degradative ubiquitin ligases TRAF6 and linear ubiquitin chain assembly complex (LUBAC) consisting of HOIP, HOIL1, and Sharpin (Kirisako *et al*, 2006; Gerlach *et al*, 2011; Ikeda *et al*, 2011; Tokunaga *et al*, 2011) in TBK1/IKKε DKO cells (Fig 3A–C). The observation that LUBAC is reproducibly detectable only upon TBK1/IKKε ablation was intriguing, since LUBAC functions as a potent activator of NF-κB in several proinflammatory complexes, such as TNF or IL-1α (Hrdinka & Gyrd-Hansen, 2017). Ubiquitin ligase activity of TRAF6 and LUBAC create K63- and M1-ubiquitin linkages, respectively, which provide binding platforms for adaptors TAB 1/2/3 and

associated kinase TAK1 and NEMO that recruits kinases IKKα and IKKβ (Kupka *et al*, 2016). In accord, the recruitment of all these proteins to IL-17RSC was enhanced in TBK1/IKKε DKO cells (Figs 3A and EV3B). Interestingly, the recruitment of core adaptor ACT1 was not affected by the absence of both kinases, although ACT1 was not phosphorylated in TBK1/IKKε DKO cells (Fig 3A–C). On the other hand, proteins TANK and NAP1, two adaptors, constitutively associated with TBK1 and IKKε (Fig EV3C and D) were absent, while degradative ubiquitin ligase complex βTrCP1/2-Cullin1 and TRAF2 recruitment was diminished in TBK1/IKKε DKO cells (Figs 3A and B, and EV3B).

Because LUBAC subunits and TAB/TAK1 complex were detected in IL-17RSC upon ablation of TBK1/IKKε, we addressed their potential roles in the IL-17 signaling. Inhibition of TAK1 blocked the IL-17-dependent activation of JNK and p38 both in the presence or absence of the TBK1/IKKε inhibitor, showing that TAK1 is indispensable for triggering the MAPK pathway in IL-17 signaling (Fig 3D). Likewise, the LUBAC-deficient HOIP KO cells showed substantially impaired activation of NF-κB both in the presence and absence of the TBK1/IKKε inhibitor (Fig 3E) and reconstitution of HOIP KO cells with HOIP(WT), but not an empty vector, rescued IL-17 signaling (Fig EV3E). These data show for the first time that LUBAC is a component of IL-17RSC promoting the activation of NF-κB. Overall, the data demonstrate that TBK1/IKKε-mediated inhibition of IL-17 proximal signaling is mediated by restriction of the recruitment of ubiquitin ligases TRAF6, LUBAC, and effector kinases TAK1 and IKKα/β to the IL-17RSC.

While inhibition of TBK1/IKKε led to markedly enhanced activation of signaling and transcriptional responses (Figs 2 and EV2), cells deficient in TBK1/IKKε did promote gene-activating signaling pathways, but were unable to induce accumulation of target cytokines mRNA (Fig EV3F). The most probable explanation for this apparent discrepancy is the differential recruitment of TRAF2 into the IL-17RSC in these two scenarios. Whereas TRAF2 was depleted from the IL-17RSC in the complete absence of TBK1/IKKε (Fig 3B), chemical inhibition of TBK1/IKKε kinase activity augmented the recruitment of TRAF2 to the complex (Fig 3F). The major role of TBK1/IKKε presence for TRAF2 recruitment in IL-17RSC was intriguing. TRAF2 is not required for IL-17-induced proximal

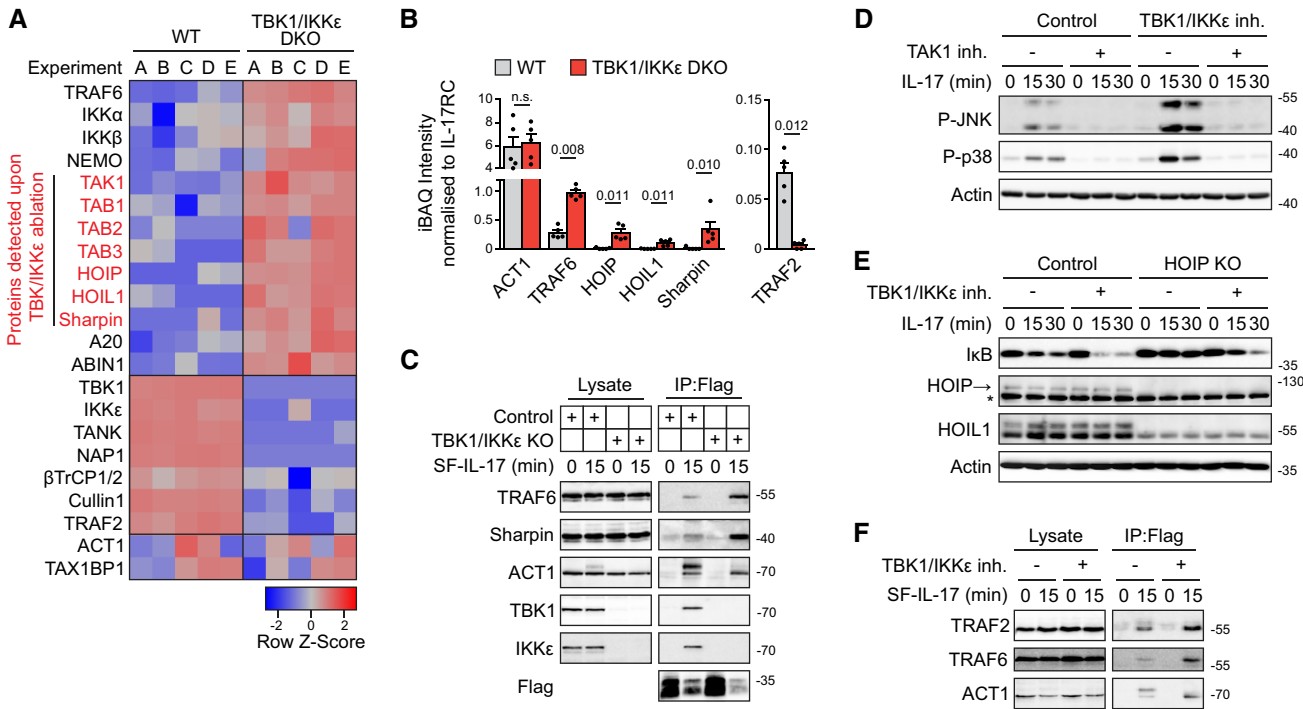

**Figure 3. Kinases TBK1 and IKKε modulate the composition of IL-17RSC.**

A, B ST2 wild-type or TBK1/IKKε DKO cells were stimulated for 15 min with SF-IL-17 (500 ng/ml) and solubilized and IL-17RSC was isolated via consecutive Flag and Strep immunoprecipitation and analyzed by MS. (A) The heat map shows the row-normalized Z-Score calculated from log₂ transformed iBAQ intensities from five independent experiments. (B) The ratio between iBAQ intensities of selected IL-17RSC components to iBAQ intensity of IL-17RC. Mean + SEM from five independent experiments is shown, and statistical significance was determined by two-tailed Mann-Whitney test.

C ST2 wild-type or TBK1/IKKε DKO cells were stimulated with SF-IL-17 (500 ng/ml) for 15 min or were left unstimulated and IL-17 was added post-lysis. Lysates were subjected to anti-Flag immunoprecipitation to isolate IL-17RSC and samples were analyzed by immunoblotting. ACT1 phosphorylation detected as upper band in WT cells is absent in TBK1/IKKε DKO cells.

D Cells were pretreated with TAK1 inhibitor 7-oxozeanol (2 μM) and/or TBK1/IKKε inhibitor MRT67307 (2 μM), stimulated with IL-17 (500 ng/ml) for indicated time points and lysates were analyzed by immunoblotting.

E ST2 wild-type or HOIP-deficient cells were pretreated with TBK1/IKKε inhibitor MRT67307 (2 μM), stimulated with IL-17 (500 ng/ml) as indicated, solubilized, and analyzed by immunoblotting. *indicates unspecific band.

F ST2 cells pretreated or not with TBK1/IKKε inhibitor MRT67307 were stimulated with SF-IL-17 for 15 min or were left unstimulated and IL-17 was added post-lysis. IL-17RSC was isolated and analyzed via immunoblotting.

Data information: Immunoblot results are representative of two (F), three (C, D) or four (E) independent experiments.
Source data are available online for this figure.

---

signaling; however, it promotes stabilization of target transcripts via several proteins regulating mRNA stability (Sun *et al*, 2011; Herjan *et al*, 2013, 2018; Somma *et al*, 2015; Amatya *et al*, 2018).

Altogether, our data show that TBK1/IKKε kinases have a dual role in IL-17RSC function: (i) Enzymatic activity of TBK1/IKKε leads to the inhibition of TRAF6 recruitment and markedly decreased MAPK and NF-κB signaling, while (ii) kinase activity-independent adaptor role of TBK1/IKKε promotes recruitment of TRAF2, a regulator of mRNA stability of proinflammatory cytokines (Swaidani *et al*, 2019).

**TBK1 and IKKε are recruited to TRAF6-generated ubiquitin linkages via NEMO**

To elucidate the molecular mechanism underlying the TBK1 and IKKε function, we addressed how these kinases are recruited into the IL-17RSC. First, we observed that cells deficient in ACT1, the

very proximal IL-17R-interacting protein (Chang *et al*, 2006; Qian *et al*, 2007), show completely disrupted IL-17RSC formation (Fig 4A and Table EV4) and disabled the activation of TBK1 and IKKε upon IL-17 stimulation (Fig EV4A). The main role of ACT1 is to recruit and activate a non-degradative E3 ubiquitin ligase TRAF6 (Schwandner *et al*, 2000). Accordingly, the presence of both ACT1 and TRAF6 is crucial for the recruitment of TBK1, IKKε, and associated adaptors TANK and NAP1 to IL-17RSC and activation of both kinases (Figs 4B and EV4B). To elucidate whether TRAF6 enzymatic activity is necessary for the recruitment of these molecules, we reconstituted TRAF6 KO cells with either wild-type TRAF6(WT) or enzymatically inactive TRAF6(C70A) mutant and analyzed the composition of IL-17RSC via MS. The formation of the IL-17RSC was largely disrupted in cells expressing the enzymatically inactive TRAF6(C70A) including the recruitment and activation of TBK1 and IKKε. We also noted that TRAF6 enzymatic activity was necessary for the recruitment of the inhibitory deubiquitinase A20 (Figs 4C

and D, and EV4C and Table EV4). A20 was previously shown to modulate IL-17 signaling, but was proposed to bind directly to IL-17RA (Garg *et al*, 2013). However, our data imply that A20 complex is recruited via non-degradative polyubiquitin linkages. Interestingly, TRAF6(C70A) recruitment to IL-17RSC was markedly enhanced in comparison with TRAF6(WT) (Fig 4D), further demonstrating that IL-17RSC assembly is regulated by potent negative feedback loop that is triggered downstream of TRAF6 activation.

The recruitment of TBK1 and IKKε to TNF-RSC requires linear ubiquitin linkages synthetized by LUBAC (Lafont *et al*, 2018; Xu *et al*, 2018). In contrast, our data show that LUBAC is only very weakly recruited to IL-17RSC and the knockout of HOIP, the main enzymatic subunit of LUBAC, did not prevent the activation of TBK1 and IKKε via IL-17 (Fig EV4D). These data show that TRAF6, but not LUBAC, activity is indispensable for TBK1 and IKKε activation via IL-17. Indeed, cells lacking the TRAF6 enzymatic activity are completely unresponsive to IL-17 even in the presence of TBK1 and IKKε inhibitor (Fig 4E and F).

NEMO is crucial for the recruitment of IKKα and IKKβ to a variety of signaling complexes. However, it was also shown to recruit TANK and NAP1 to the M1-ubiquitin chains in the TNF-RSC (Chariot *et al*, 2002; Lafont *et al*, 2018). Since NEMO binds to TRAF6-mediated K63-ubiquitin as well (Ea *et al*, 2006; Wu *et al*, 2006; Laplantine *et al*, 2009), we hypothesized that NEMO recruits TANK and NAP1 adaptors bound to TBK1 and IKKε to the IL-17RSC. Indeed, NEMO-deficient cells failed to efficiently activate TBK1 and IKKε upon IL-17 stimulation (Fig 4G). Accordingly, the recruitment of these kinases and their activation within IL-17RSC was reduced by approximately 80% in the absence of NEMO in both murine and human cell lines as revealed by the quantification of Western blots (Figs 4H and I, and EV4E and F). Activation of TBK1/IKKε upon IL-17 signaling was rescued in NEMO KO cells reconstituted with NEMO(WT), but not NEMO(Δ201–248) lacking TANK/NAP1-binding domains (Chariot *et al*, 2002; Lafont *et al*, 2018) or an empty vector (Fig EV4G). Altogether, NEMO is the major protein connecting TBK1 and IKKε with K63-linkages formed by TRAF6.

TBK1 and IKKε activation requires phosphorylation of Ser172 located in the kinase domain activation loop, which can be mediated either by TBK1/IKKε themselves or by IKKα/β (Ma *et al*, 2012; Larabi *et al*, 2013). In accord, only combined inhibition of IKKα/β and TBK1/IKKε kinases prevented TBK1/IKKε phosphorylation, while inhibition of IKKα/β alone or their upstream kinase TAK1 had no apparent effect (Fig EV4H). These results provide further evidence that NEMO-associated kinases IKKα/β might contribute to, but are not essential for, TBK1/IKKε activation.

### NEMO has unprecedented dual role in the IL-17 signaling

An interesting prediction of our model of proximal IL-17 signaling is that NEMO has a dual role in the IL-17 signaling as it acts both as a positive regulator by recruiting IKKα/β effector kinases and a negative regulator by recruiting TBK1/IKKε to the IL-17RSC. To test this hypothesis, we compared the IL-17RSC composition in WT and NEMO KO cells via MS (Table EV5). NEMO deficiency dramatically changed the composition of IL-17RSC (Fig EV5A). In NEMO KO cells, the kinases IKKα and IKKβ were absent from IL-17RSC, while the recruitment of TBK1 and IKKε was substantially decreased. Moreover, these cells exhibited strongly enhanced recruitment of

TRAF6 (8.4-fold) and TAB/TAK1 complex to the IL-17RSC (Figs 5A–C and EV5B). In contrast, ACT1 recruitment was only very slightly increased (1.2-fold), although its phosphorylation was not detectable (Fig 5B and C). Reconstitution of NEMO KO cells with NEMO(WT) enabled the recruitment of phosphorylated TBK1 and IKKε to the IL-17RSC, leading to the phosphorylation of ACT1 and decreased recruitment of TRAF6, similarly to WT cells. In contrast, NEMO(Δ201–248) lacking the TANK/NAP1 interaction site was unable to rescue the phenotype (Fig EV5C).

Subsequently, we analyzed the IL-17 signaling pathways in WT and NEMO KO cells. Whereas the NF-κB pathway was abolished, the activation of MAPKs, JNK, and p38 was strongly enhanced in NEMO KO cells compared to WT cells (Fig 5D). In accord, the reconstitution of NEMO KO cells with NEMO (WT), but not with NEMO (Δ201–248) or an empty vector, inhibited the activation of the MAPK pathway by IL-17 (Fig EV5D). Moreover, the inhibition of TBK1 and IKKε had no effect in NEMO-deficient cells, demonstrating that NEMO is critical for the function of these kinases in IL-17 signaling (Fig EV5E). We confirmed that NEMO negatively regulates the recruitment of TRAF6 into the IL-17RSC and the IL-17-triggered MAPK activation also in human HeLa cells (Fig EV5F–H).

The negative role of NEMO in the IL-17-induced MAPK activation was independent of its well established role in the IKKα and IKKβ activation, because the chemical inhibition of these two kinases had no effect on the JNK and p38 activation (Fig 5E). Importantly, NEMO deficiency does not enhance the activation of JNK and p38 upon stimulation with TNF (Fig 5F). Altogether, our experiments identified a unique negative feedback loop in the proximal IL-17 signaling pathway, represented by NEMO-mediated recruitment of TBK1 and IKKε to the IL17-RSC causing a release of the key ubiquitin ligase TRAF6 from the complex (Fig 5G).

### Phosphorylation of ACT1 by NEMO-recruited TBK1 and IKKε on multiple residues inhibits TRAF6 recruitment

The absence of NEMO or both TBK1 and IKKε does not affect the recruitment of ACT1, but increases the amount of TRAF6 in the IL-17RSC both in murine and human cells (Figs 3–5 and EV6A). Accordingly, the chemical inhibition of TBK1 and IKKε enhanced TRAF6 recruitment to the IL-17RSC (Figs 3F and EV6B). TBK1/IKKε activity correlates with ACT1 phosphorylation upon IL-17 stimulation (Figs 3C, 4B, 4D, and 5C and EV5C). ACT1 was shown to be degraded upon prolonged IL-17 stimulation via Cullin1-dependent mechanism (Shi *et al*, 2011). Although our data show that ablation of TBK1/IKKε leads to decreased recruitment of Cullin1 to IL-17RSC (Figs 3A and EV3B), we did not observe increased phosphorylation of p38 and JNK in Cullin1 KO cells upon IL-17 stimulation, even though Cullin1-mediated degradation of IκB was inhibited (Fig EV6C and D). Therefore, a different mechanism inhibiting TRAF6 recruitment to IL-17RSC must be responsible for the inhibitory effect of TBK1/IKKε-mediated ACT1 phosphorylation.

ACT1 associates with IL-17R via its C-terminal SEFIR domain (Chang *et al*, 2006; Liu *et al*, 2011) and interacts with TRAF6 via its first 15 N-terminal amino acids (Sonder *et al*, 2011) (Fig 6A). In accord, the reconstitution of ACT1-deficient cells with a series of deletion mutants demonstrated that deletion of first 20 amino acids of ACT1 completely prevented IL-17 signaling (Fig EV6E). Interestingly, the analysis of ACT1 structure *in silico* (Mizianty *et al*, 2013;

Hanson *et al*, 2017; Meszaros *et al*, 2018) indicated that the mid-part separating TRAF6-binding site and the SEFIR domain is highly disordered (Fig 6B). Our MS data revealed that ACT1 in the IL-17RSC was phosphorylated at multiple sites, all of them being within the unstructured mid-part (Fig 6A and Table EV6). Furthermore, none of the phospho-sites is located in the close proximity to the TRAF6-binding domain, raising a question how these

phosphorylation events affect TRAF6 recruitment. The reconstitution of ACT1-deficient cells with ACT1 protein mutated in individual phospho-sites to alanines did not affect the signaling outcome (Fig 6C), suggesting that there is not a single critical phospho-site in ACT1. However, mutation of all these phospho-sites substantially increased signaling responses to IL-17 (Fig 6D) and strongly enhanced recruitment of TRAF6 to IL-17RSC (Fig 6E). Altogether,

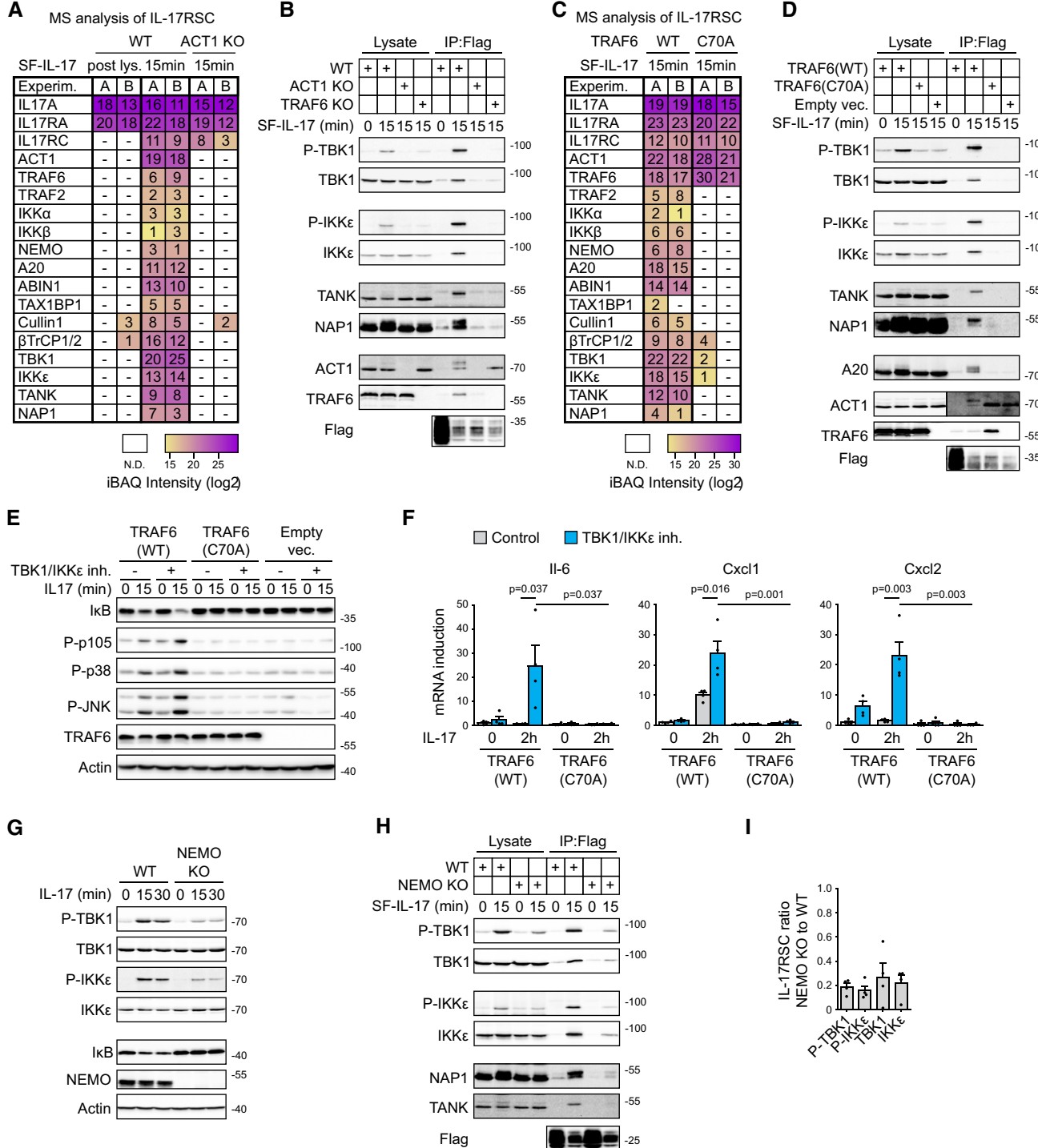

Figure 4.

◀

**Figure 4.  Enzymatic activity of TRAF6 and NEMO is crucial for the recruitment of both TBK1 and IKKε to IL-17RSC.**

A    ST2 wild-type and ACT1-deficient cells were stimulated for 15 min with SF-IL-17 (500 ng/ml) and solubilized and IL-17RSC was isolated via consecutive Flag and Strep immunoprecipitation and analyzed by MS. As a control, cells were first solubilized and SF-IL-17 was added only post-lysis. Number of identified peptides (unique + razor) and iBAQ intensities for each protein in two independent experiments are shown.

B    ST2 wild-type, ACT1 KO or TRAF6 KO cells were stimulated with SF-IL-17 (500 ng/ml) as indicated or were left unstimulated and IL-17 was added post-lysis. Lysates were subjected to anti-Flag immunoprecipitation to isolate IL-17RSC and samples were analyzed by immunoblotting.

C    ST2 cells deficient in TRAF6 reconstituted with TRAF6(WT) or enzymatically inactive TRAF6(C70A) were stimulated for 15 min with SF-IL-17 (500 ng/ml), solubilized and IL-17RSC was isolated via consecutive Flag and Strep immunoprecipitation and analyzed by MS. Number of identified peptides (unique + razor) and iBAQ intensities for each protein in two independent experiments are shown.

D    TRAF6-deficient cells reconstituted with TRAF6(WT), TRAF6(C70A), or empty vector were stimulated with SF-IL-17 (500 ng/ml) as indicated or were left unstimulated and IL-17 was added post-lysis. IL-17RSC was isolated from lysates by anti-Flag immunoprecipitation and analyzed by immunoblotting. Short and long exposure of the same membrane stained for Act1 are shown for lysate and IP samples, respectively.

E, F TRAF6-deficient cells reconstituted with the indicated constructs were pretreated or not with TBK1/IKKε inhibitor MRT67307 (2 μM) and stimulated with IL-17 (500 ng/ml) for indicated time. (E) The activation of signaling pathways analyzed by immunoblotting. (F) The induction of mRNA for selected genes analyzed by real-time PCR. Mean + SEM from four independent experiments is shown, and statistical significance was determined using unpaired two-tailed Student's *t*-test.

G    ST2 cells wild-type or NEMO KO were stimulated with IL-17 (500 ng/ml) for indicated time points and solubilized and analyzed by immunoblotting.

H, I  ST2 wild-type or NEMO KO cells were stimulated with SF-IL-17 (500 ng/ml) as indicated or were left unstimulated and IL-17 was added post-lysis. Lysates were subjected to anti-Flag immunoprecipitation to isolate IL-17RSC. Samples were analyzed by immunoblotting (H). In addition, results from four independent experiments were quantified by densitometry and mean + SEM from four independent experiments is shown (I).

Data information: Immunoblot results are representative of two (B, E), three (D), or four (G–I) independent experiments.
Source data are available online for this figure.

these data showed that TBK1 and IKKε phosphorylate ACT1 at multiple sites in the disordered mid-part of the protein to limit the amount of recruited TRAF6. In accord, reconstitution of ACT1 KO cells with ACT1 (Δ20–380) deletion mutant led to markedly enhanced proximal signaling as compared to ACT1(WT) (Fig 6F).

Based on these data, we propose a new model of IL-17RSC formation and regulation. Triggering of the IL-17RA/IL-17RC by dimeric IL-17 leads to the recruitment of approximately six ACT1 molecules as evident from our MS analysis (Fig 6G). These accumulated ACT1 molecules provide multiple TRAF6-binding sites, generating a high-avidity docking site for TRAF6 trimers (Ye *et al*, 2002). TRAF6-mediated K63-ubiquitin chains promote the recruitment and activation of effector molecules and kinases TBK1 and IKKε via NEMO. TBK1/IKKε have dual role: on one hand, they recruit TRAF2 in a kinase activity-independent manner that is not required for activation of proximal signaling pathways, but promotes expression of target genes by stabilization of their transcripts. On the other hand, TBK1 and IKKε phosphorylate ACT1 at multiple sites within its unstructured mid-part separating SEFIR- and TRAF6-binding domains. This results in a strong negative charge-mediated repulsion between individual ACT1 molecules. Subsequently, individual TRAF6-binding domains are separated leading to the release of TRAF6 trimers from the complex (Fig 7). In this manner, NEMO-mediated recruitment of TBK1 and IKKε kinases provide a potent negative feedback loop, which limits the IL-17 signaling pathway.

## Discussion

Proper regulation of IL-17RSC assembly and signaling is crucial for efficient immune responses while preventing autoimmunity. Our study provides a new methodological approach to analyze the assembly of the early signaling complex formed immediately upon binding of IL-17 to its receptors. Based on this methodology, we identified two kinases, TBK1 and IKKε, as major components of the IL-17RSC, whose recruitment is over 50-fold increased when compared with related kinases IKKα and IKKβ. We demonstrated that these kinases are very strongly activated upon IL-17

stimulation, to levels comparable with strongly proinflammatory cytokines IL-1α or TNF.

The role of these kinases in IL-17 signaling has been very controversial. Two highly influential studies showed that the ablation of either TBK1 or IKKε alone inhibited IL-17 transcriptional responses (Bulek *et al*, 2011; Herjan *et al*, 2018). These articles promoted the view that these kinases have an activation role (Monin & Gaffen, 2018; Li *et al*, 2019; McGeachy *et al*, 2019; Swaidani *et al*, 2019). In accord, a recent publication analyzing the transcriptional response of cells upon IL-17 stimulation in the presence of TNF demonstrated that deficiency of TBK1/IKKε led to markedly low activation of transcription, again supporting the view that these kinases propagate IL-17 signaling (Tanaka *et al*, 2019). However, cells deficient in TBK1/IKKε are highly prone to TNF-induced cell death (Bonnard *et al*, 2000; Matsui *et al*, 2006; Lafont *et al*, 2018; Xu *et al*, 2018) and it has been shown that mouse embryonic fibroblast deficient in TBK1/IKKε, used by Tanaka *et al*, rapidly succumb to TNF-induced apoptosis and necroptosis (Lafont *et al*, 2018). Therefore, the irresponsiveness of TBK1/IKKε DKO cells to a combined IL-17 and TNF treatment can be explained by their death.

On the other hand, another study described that overexpression of TBK1 or IKKε in TBK1/IKKε DKO cells inhibits the IL-17 signaling via reducing the recruitment of TRAF6 to ACT1 (Qu *et al*, 2012). However, these experiments relied on overexpression systems and the molecular mechanism explaining the observed phenotype was only partially uncovered. Perhaps, this is why this study is either ignored or misinterpreted in a number of recent reviews on the topic of IL-17 signaling (Gaffen *et al*, 2014; Monin & Gaffen, 2018; Li *et al*, 2019; McGeachy *et al*, 2019; Swaidani *et al*, 2019). In this study, we resolved the controversy and developed a model explaining the role of TBK1 and IKKε in regulation of IL-17RSC assembly and signaling.

The core of IL-17RSC contains dimeric IL-17, one IL-17RA and one IL-17RC subunits (Ely *et al*, 2009; Liu *et al*, 2013; Goepfert *et al*, 2017). Our data demonstrated that this complex then recruits on average six ACT1 molecules. The molecular basis for ACT1 enrichment is presently unclear. ACT1 was described to form homo-oligomers (Mauro *et al*, 2003; Liu *et al*, 2011; Boisson *et al*, 2013) and it is possible that once ACT1 is recruited, it can provide docking sites for

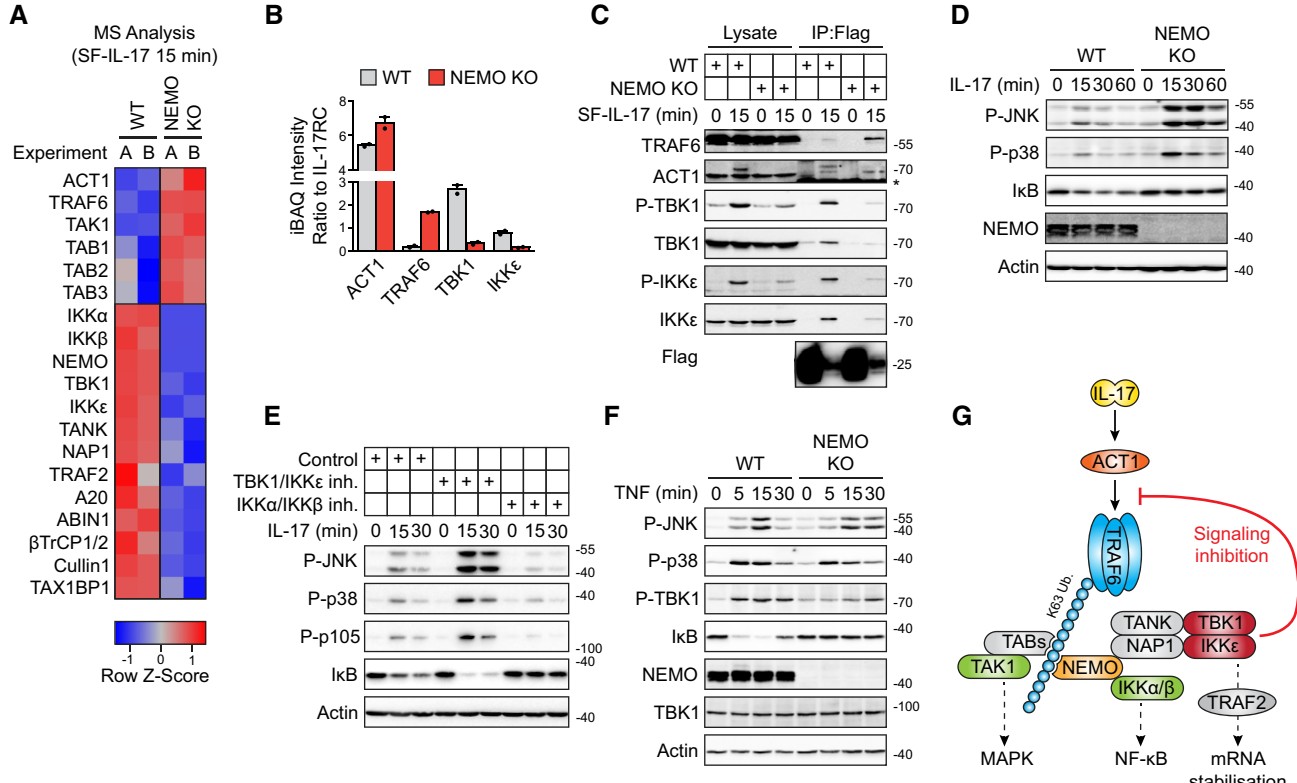

**Figure 5.  NEMO functions as an inhibitor of IL-17RSC assembly.**

A, B   ST2 wild-type or NEMO KO cells were stimulated for 15 min with SF-IL-17 (500 ng/ml), solubilized and IL-17RSC was isolated via consecutive Flag and Strep immunoprecipitation and analyzed by MS. (A) The heat map shows the row-normalized Z-Score calculated from $\log_2$ transformed iBAQ values from two independent experiments. (B) The ratio between iBAQ intensities of selected IL-17RSC components to iBAQ intensity of IL-17RC. Mean + SEM from two independent experiments is shown.

C, D   ST2 wild-type or NEMO KO cells were stimulated with SF-IL-17 (500 ng/ml) as indicated or were left unstimulated and IL-17 was added post-lysis. Lysates were subjected to anti-Flag immunoprecipitation to isolate IL-17RSC (C) or tested for activation of signaling pathways (D) and analyzed by immunoblotting.

E   ST2 cells were pretreated with TBK1/IKKε inhibitor MRT67307 (2 μM) or IKKα/IKKβ inhibitor TPCA1 (10 μM), stimulated with IL-17 (500 ng/ml) as indicated and analyzed by immunoblotting.

F   ST2 wild-type or NEMO KO cells were stimulated with TNF (50 ng/ml) for indicated time points and lysates were analyzed by immunoblotting.

G   Schematic model of the negative feedback loop mediated by NEMO-recruited TBK1/IKKε.

Immunoblot results are representative of four (C, D), three (E), or two (F) independent experiments.

Source data are available online for this figure.

additional ACT1 molecules. The enrichment of ACT1 proteins in the complex provides multiple binding sites for trimeric K63-ubiquitin ligase TRAF6. Importantly, TRAF6 has relatively low affinity to monomeric scaffolds (Ye *et al*, 2002) and oligomerization of ACT1 likely enables high-avidity interactions, which explains the molecular mechanism of TRAF6 enrichment at the IL-17RSC. The recruitment of TRAF6 seems to be absolutely critical for IL-17RSC assembly, since the ablation of TRAF6 or its enzymatic activity prevented recruitment of all other components of the complex. Analysis of various complexes dependent on formation of K63-linkages for signaling showed a similar pattern of molecules that are recruited via binding to polyubiquitin linkages, including signaling complexes like TAB/TAK1 and NEMO/IKKα/IKKβ (Zinngrebe *et al*, 2014; Shimizu *et al*, 2015). However, these molecules are recruited very weakly to IL-17, which correlates with poor signaling response of cells upon IL-17 stimulation.

The canonical role of NEMO in IL-17 and other signaling pathways is the recruitment and activation of IKKα/IKKβ kinases to

trigger NF-κB pathway (Hinz & Scheidereit, 2014). Here, we revealed an unexpected dual role of K63-ubiquitin binding adaptor NEMO in IL-17 signaling. NEMO recruits adaptors TANK and NAP1 which in turn bring two closely related kinases TBK1 and IKKε to IL-17RSC, similarly to its role in TNF-RSC (Clark *et al*, 2011b; Lafont *et al*, 2018). TBK1/IKKε are recruited to various signaling complexes, and depending on the particular context, they can modulate various cellular processes, ranging from the induction of mitophagy and autophagy (Wild *et al*, 2011; Heo *et al*, 2015; Thurston *et al*, 2016), interferon signaling (Fitzgerald *et al*, 2003; Sharma *et al*, 2003; Perry *et al*, 2004) to the protection from TNF-induced cell death (Bonnard *et al*, 2000; Lafont *et al*, 2018; Xu *et al*, 2018).

Our data revealed that in the case of IL-17RSC assembly, TBK1 and IKKε trigger a robust negative feedback loop by phosphorylating ACT1 on a number of residues in a long unstructured stretch of amino acids separating SEFIR domain and the TRAF6-binding site. The sites identified in this study are partially different from previous

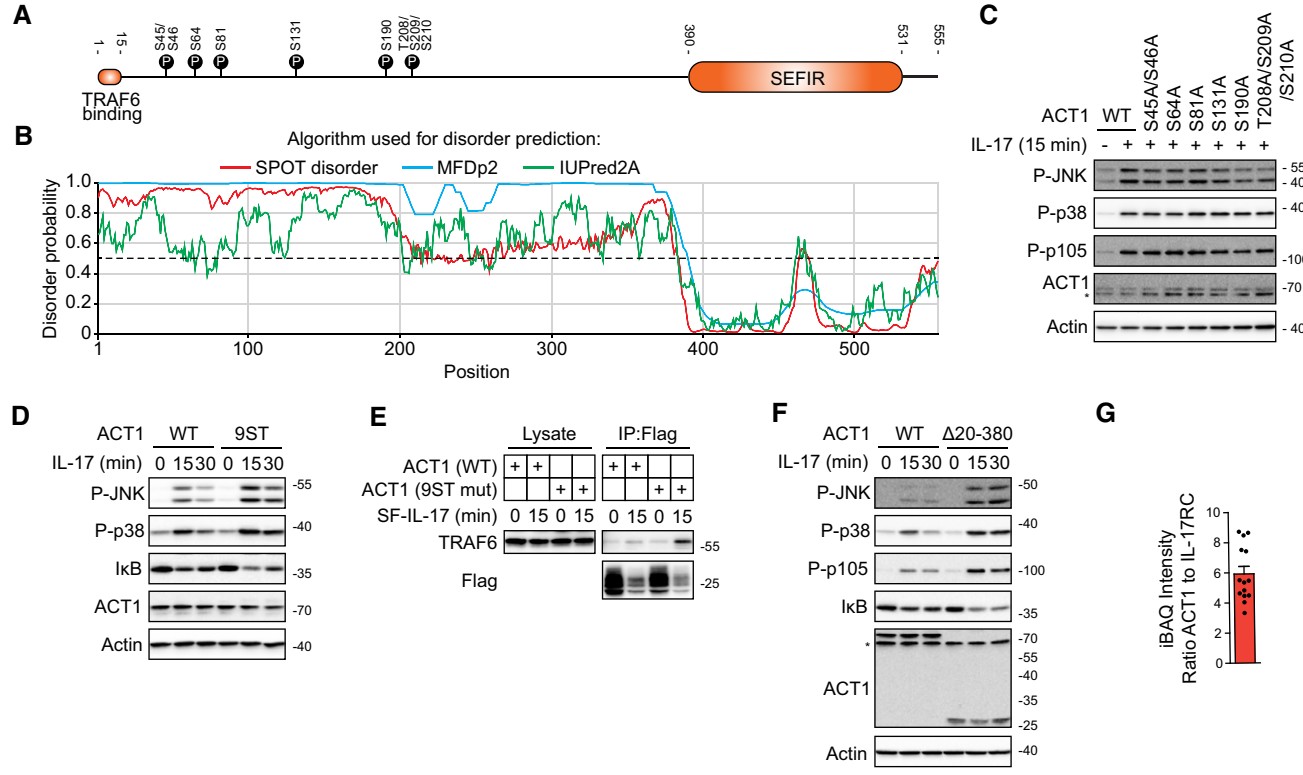

**Figure 6. Phosphorylation of ACT1 in disordered region leads to the release of TRAF6 from IL-17RSC.**

A    Schematic representation of ACT1 with indicated TRAF6-binding domain, SEFIR domain and location of phosphorylation sites identified in our MS analysis (based on Table EV6).

B    Analysis of ACT1 structure using three different algorithms predicting disordered regions. Detected phosphorylation sites are all in the disordered region of the protein.

C, D    ACT1 KO cells were reconstituted with either wild-type ACT1 or indicated mutants or ACT1 with all nine identified phospho-sites mutated to alanines (9ST mut). Cells were stimulated with IL-17 (500 ng/ml) as indicated and lysates were analyzed by immunoblotting. *indicates nonspecific band.

E    ACT1 KO cells expressing either ACT1(WT) or ACT1(9ST mut) were stimulated with SF-IL-17 (500 ng/ml) as indicated or were left unstimulated and IL-17 was added post-lysis. IL-17RSC was isolated via anti-Flag immunoprecipitation and analyzed by immunoblotting.

F    ACT1 KO cells were reconstituted with either ACT1(WT) or ACT1 (Δ20–380). Cells were stimulated with IL-17 (500 ng/ml) as indicated and lysates were analyzed by immunoblotting. *indicates nonspecific band.

G    Analysis of iBAQ intensity of ACT1 normalized to iBAQ intensity of IL-17RC. Data are composite of all 14 MS analyses of IL-17RSC from wild-type ST2 cells performed in this study. Mean + SEM is shown.

Data information: Immunoblot results are representative of two (C, E) or four (D, F) independent experiments.
Source data are available online for this figure.

reports (Bulek *et al*, 2011; Qu *et al*, 2012), which might reflect either differences in the experimental setup or in the cell lines used. However, our data indicate that phosphorylation of any of the phosphorylation sites within this part of ACT1 contributes to the spatial separation of TRAF6-binding sites, leading to decreased avidity of TRAF6 for ACT1 oligomers and its release from the IL-17RSC. This mechanism limits the ability of TRAF6 to create K63-ubiquitin linkages and promote recruitment of additional effector molecules, but still ensures that some basal signaling is triggered. The inhibitory role of NEMO mediated by the recruitment of TBK1/IKKε is unprecedented as it has not been described previously and it currently seems to be unique for IL-17. The ablation of the TBK1/IKKε-mediated negative feedback loop completely changes the composition of the receptor by enhancing the recruitment of ubiquitin-binding proteins, which allows the detection of signaling components such as LUBAC and TAB/TAK1. LUBAC amplifies signaling by recruiting additional NEMO/IKKα/IKKβ and enhancing NF-κB

activation (Kirisako *et al*, 2006; Haas *et al*, 2009; Tokunaga *et al*, 2009; Hrdinka & Gyrd-Hansen, 2017), which is reflected in markedly enhanced transcription of IL-17 target genes.

At the same time, TBK1/IKKε recruit TRAF2 in a kinase activity-independent manner, which is required for the expression of IL-17 responsive genes via stabilization of their transcripts (Sun *et al*, 2011; Herjan *et al*, 2013, 2018; Somma *et al*, 2015; Amatya *et al*, 2018). The exact molecular mechanism of how TRAF2 is recruited to the IL-17RSC remains to be resolved, although both TBK1 and IKKε were shown previously to associate with TRAF2 (Pomerantz & Baltimore, 1999; Bonnard *et al*, 2000; Shen *et al*, 2012). Interestingly, we did not detect TRAF5, another reported regulator of mRNA stability (Sun *et al*, 2011), in the IL-17RSC. One possibility is that the recruitment of TRAF5 to the complex might be weak or transient and below the detection limit of our method. Altogether, TBK1/IKKε kinases are major regulators of the complex and modulate its function. They inhibit recruitment of TRAF6 and therefore induction of

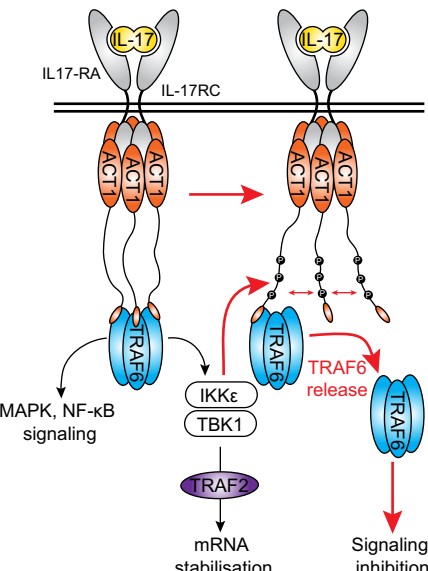

**Figure 7. Schematic representation of the negative feedback loop modulating the assembly of IL-17RSC.**

Upon binding of IL-17 to IL-17RA and IL-17RC, 6 molecules of ACT1 are recruited, creating docking site for trimeric TRAF6. TRAF6-created K63-polyubiquitin linkages promote activation of signaling pathways, but also trigger recruitment of TBK1 and IKKε kinases to IL-17RSC. Both kinases then phosphorylate ACT1 in disordered region of the protein. This leads to a spatial separation of TRAF6-binding sites present on ACT1, decreased avidity for TRAF6 and its release from IL-17RSC, ultimately leading to inhibition of signaling. In addition, TBK1/IKKε enables recruitment of TRAF2 in kinase activity-independent manner to promote stabilization of mRNA of target cytokines.

signaling, while they promote recruitment of TRAF2 to promote expression of IL-17 responsive genes. These results reconcile various seemingly contradictory findings concerning the role of TBK1/IKKε as both activators and inhibitors of signaling. The present biochemical data were not yet confirmed using *in vivo* animal models. However, the results presented in this work showing that inhibition of TBK1/IKKε leads to massively enhanced IL-17-induced signaling in both human and murine cell lines argue against potential therapeutic targeting of TBK1/IKKε kinase activity in treatment of IL-17 mediated autoimmune diseases.

Our data identified a molecular mechanism for activation and regulation of IL-17 signaling. The NEMO-TBK1/IKKε-mediated negative feedback loop allows only weak transcriptional response to IL-17 stimulation. Given the major role of IL-17 in orchestrating the function of the immune system, the tight control of IL-17-induced signaling via the NEMO-TBK1/IKKε axis contributes to balancing the efficient immune response to pathogens and self-tolerance.

# Materials and Methods

## Cell lines, reagents, and antibodies

ST2 cells were kindly provided by Jana Balounova (Institute of Molecular Genetics, Prague). HeLa, Hek293, and ØNX-Eco, and ØNX-Ampho cells were kindly provided by Tomas Brdicka (Institute of Molecular Genetics, Prague). Cells were cultivated in DMEM

supplemented with 10% fetal calf serum (FCS; Gibco), 100 U/ml penicillin (BB Pharma), 100 μg/ml streptomycin (Sigma-Aldrich), and 40 μg/ml gentamicin (Sandoz). Cells were regularly tested for mycoplasma contamination. Murine and human IL-1α and murine TNF were purchased from Peprotech. Human TNF was produced as described previously (Draber *et al*, 2015). Inhibitors MRT67307, 7-oxozeanol, and TPCA-1 were from Tocris.

Primary antibodies used in this study were purchased from Cell Signaling: TBK1 (Cat#3013), P-TBK1(S172) (D52C2, Cat#5483), murine IKKε (D61F9, Cat#3416), human IKKε (D20G4, Cat#2905), P-IKKε (S172) (D1B7, Cat#8766), P-IKKα/β (S176/180) (16A6, Cat#2697), IκB (Cat#9242), P-IκB (Ser32/36) (5A5, Cat#9246), P-p105 (Ser933) (18E6, Cat#4806), P-p38 (Thr180/Tyr182) (D3F9, Cat#4511), P-JNK (Thr183/Tyr185) (98F2, Cat#4671), P-ATF2 (Thr71) (Cat#9221), P-p65 (Ser536) (93H1, Cat#3033), TANK (Cat#2141), Myc (9B11, Cat#2276), TRAF2(C192, Cat#4724); from Abcam: NAP1/AZI2 (EPR14698, Cat#ab192253), TRAF6 (EP592Y, Cat#ab40675); from Santa Cruz: ACT1 (D-11, Cat#sc-398161), NEMO (FL-419, Cat#sc-8330); from Sigma: Flag (M2, Cat#F3165), Actin (AC-15, Cat#A1978); from MRC PPU: HOIP (S174D); HOIL1 (S150D); from ProteinTech: Sharpin (Cat#14626-I-AP), and from BioLegend: Flag-APC (L5, Cat#637307).

## Generation of knockout cell lines

In order to generate knockout cell lines via CRISPR/Cas9 approach, single-guided RNA (sgRNA) targeting selected genes were designed using web tool CHOPCHOP (Labun *et al*, 2019) and inserted in pSpCas9(BB)-2A-GFP (PX458) vector kindly provided by Feng Zhang (Addgene plasmid #48138) (Ran *et al*, 2013). All constructs were sequenced. Below is the list of sgRNA target sequences with highlighted PAM motif (in bold and underlined) used to knockout different murine or human proteins:
Mouse TBK1 KO: 5′-GGAAGTCCATACGCATTGGA**CGG**
Mouse IKKε KO: 5′-GGCTGGCATGAACCACCTGC**GGG**
Mouse IKKε in TBK1/IKKε DKO: 5′-GGGCCCACCGAAGGGGAT-GA**AGG**
Mouse HOIP KO: 5′-GATGGATTGAGTTTCCCCGA**AGG**
Mouse ACT1 KO: 5′-GTGGCCAAGAGATGATGCCC**CGG**
Mouse TRAF6 KO: 5′-GCGTAAAGCCATCAAGCAGAT**GGG**
Mouse NEMO KO: 5′-GTGCATTTCCAGGTCAGCCAG**CGG**
Mouse Cullin1 KO: 5′-GTGCCTACCTCAATAGACAT**TGG**
Human TBK1 KO: 5′-GGTAGTCCATAGGCATTAGA**AGG**
Human IKKε KO: 5′-GAACATCATGCGCCTCGTAG**GGG**
Human NEMO KO: 5′-GGCAGCAGATCAGGACGTAC**TGG**

Cells were transfected with PX458 vector containing particular sgRNA using Lipofectamine 2000 (Invitrogene) according to the manufacturer's instructions. Green fluorescent protein (GFP) expressing cells were sorted as single cell in 96-well plates using FACSAria IIu (BD Biosciences). The resulting colonies were screened for the expression of target proteins by immunoblotting and sequencing of DNA surrounding the sgRNA target site.

## Gene expression using retroviral vectors

Coding sequences of different proteins were inserted in retroviral pBabe vector expressing GFP marker under SV40 promoter.

Mutations in these sequences were performed by fusion PCR approach using Phusion polymerase (New England BioLabs). All constructs were sequenced.

For production of viruses, ØNX-Eco cells (used for reconstitution of ST2 cells) or ØNX-Ampho cells (used for reconstitution of HeLa cells) were transfected with pBabe-GFP vector harboring coding sequences for indicated proteins or empty vector using Lipofectamine 2000. Virus containing supernatants were collected, passed through 0.2 μl filter and added to target cells in the presence of 6 μg/ml polybrene followed by spinning 1,200 $g$ for 45 min. Infected cells were sorted as GFP positive using FACSAria IIu.

## Production and testing of recombinant SF-IL-17

DNA sequences coding the SF-IL-17 construct containing from N-terminus: CD33 leader, 6xHis, 2xStrep tag, 1xFlag tag, and either murine IL-17 (AA 26–158) or human IL-17 (AA 24–155) were prepared using GeneArt Gene Synthesis service (Thermo Fisher Scientific) and inserted into pcDNA3.1 vector. These plasmids were transfected to adherent HEK293 cells using polyethylenimine (PEI) transfection; for transfection of the cells grown on one 15-cm dish, we used 30 μg of the vector mixed with 75 μg PEI.

After 3 days, supernatants were collected and loaded on His GraviTrap TALON column (GE Healthcare) equilibrated in purification buffer (50 mM sodium phosphate pH 7.4, 300 mM NaCl). Columns were subsequently washed with 20 mM imidazole in purification buffer and eluted with 350 mM imidazole in purification buffer. In order to remove imidazol, samples were loaded on centrifugal filter (10 kDa molecular weight cutoff, Merck Millipore), washed several times with purification buffer and concentrated. The protein concentration was measured using NanoDrop (ND-1000, Thermo Fisher Scientific). Recombinant proteins were mixed with equal volume of glycerol and kept at −80°C for long-term storage.

In order to check the production of SF-IL-17, samples were mixed with SDS sample buffer, reduced with 50 mM dithiothreitol (DTT) or left untreated, separated using SDS-PAGE and gels were stained with InstantBlue Coomassie protein stain (Expedeon).

In order to test functionality of produced SF-IL-17, cells were incubated on ice with SF-IL-17 (1 μg/ml) in FACS buffer (PBS/0.2% FCS/0.1% NaN₃) and washed and binding of SF-IL-17 was detected using the APC-conjugated anti-Flag antibody on Accuri C6 cytometer (BD Biosciences) and analyzed by FlowJo (BD Biosciences).

## Cell stimulation

Prior to stimulation, ST2 or HeLa cells were washed and incubated in serum-free DMEM medium for 30–60 min. In some experiments, cells were pretreated with indicated chemical inhibitor. Depending on whether murine or human cell line was used, cells were stimulated with murine or human SF-IL-17, IL-1α, or TNF as indicated. Subsequently, cells were lysed in 1% *n*-Dodecyl-β-D-Maltoside (DDM) containing lysis buffer (30 mM Tris pH 7.4, 120 mM NaCl, 2 mM KCl, 2 mM EDTA, 10% glycerol, 10 mM chloroacetamide, 10 mM cOmplete protease inhibitor cocktail, and PhosSTOP tablets (Roche)). Samples were incubated at 4°C for 30 min, cleared by centrifugation (21,130 $g$, 30 min, 2°C), mixed with SDS sample buffer, reduced by 50 mM DTT, and analyzed by immunoblotting.

## Isolation of IL-17RSC

For each experimental condition, ST2 or HeLa cells were grown on 15-cm dish. Cells were washed and incubated 30–60 min in serum-free DMEM. In some experiments, cells were pretreated with TBK1/IKKε inhibitor MRT67307 (2 μM). Cells were stimulated in 10 ml serum-free DMEM with SF-IL-17 (500 ng/ml) for indicated time points, solubilized in 1.5 ml 1% DDM containing lysis buffer and cleared by centrifugation (21,130 $g$, 30 min, 2°C). In control samples, 0.5 μg of SF-IL-17 was added post-lysis. Cleared lysates were incubated with 10 μl of anti-FLAG M2 affinity agarose gel (Sigma) overnight, washed 3× with 0.1% DDM containing lysis buffer and eluted by boiling in SDS sample buffer with 50 mM DTT.

## Tandem affinity purification of IL-17RSC

For each experimental condition, ST2 cells were grown on 6 × 15 cm dishes. Prior to stimulation, cells were washed and incubated in serum-free DMEM for 30–60 min. Cells were activated with murine SF-IL-17 (500 ng/ml) for 15 min or left untreated. Subsequently, cells on each dish were solubilized in 1.5 ml of 1% DDM containing lysis buffer, collected, and incubated for 30 min on ice. In total, for each experimental condition 9 ml of lysates were obtained. Samples were cleared by centrifugation (21,130 $g$, 30 min, 2°C). In the control samples, 3 μg of SF-IL-17 was added post-lysis.

The first immunoprecipitation step was carried out by overnight incubation of samples with 50 μl of anti-FLAG M2 affinity agarose gel (Sigma). Subsequently, the beads were washed 3× with 0.1% DDM containing lysis buffer and isolated proteins were eluted by incubation of the beads in 250 μl of 1% DDM containing lysis buffer supplemented with 100 μg/ml of 3xFlag peptide (Sigma) overnight. The supernatant was collected and the elution step was repeated once again for 8 h.

The second purification step was carried upon incubation of the samples with 50 μl of Strep-Tactin Sepharose beads (IBA Lifesciences) overnight. The samples were subsequently washed 3× with 0.1% DDM containing lysis buffer and 1× with lysis buffer alone. Bound proteins were eluted upon incubation of the beads with 220 μl of MS Elution buffer (2% sodium deoxycholate in 50 mM Tris pH 8.5).

## Protein digestion

The eluted protein samples (200 μl) were reduced with 5 mM tris(2-carboxyethyl)phosphine at 60°C for 60 min and alkylated with 10 mM methyl methanethiosulfonate at room temperature for 10 min. Proteins were cleaved overnight with 1 μg of trypsin (Promega) at 37°C. In order to remove sodium deoxycholate, samples were acidified with 1% trifluoroacetic acid, mixed with equal volume of ethyl acetate, and centrifuged (15,700 $g$, 2 min) and aqueous phase containing peptides was collected (Masuda *et al*, 2008). This step was repeated for two more times. Peptides were desalted using in-house made stage tips packed with C18 disks (Empore) (Rappsilber *et al*, 2007) and resuspended in 20 μl of 2% acetonitrile with 1% trifluoroacetic acid.

## nLC-MS/MS analysis

The digested protein samples (12 μl) were loaded onto the trap column (Acclaim PepMap 300, C18, 5 μm, 300 Å Wide Pore, 300 μm × 5 mm) using 2% acetonitrile with 0.1% trifluoroacetic acid at a flow rate of 15 μl/min for 4 min. Subsequently, peptides were separated on Nano Reversed phase column (EASY-Spray column, 50 cm × 75 μm internal diameter, packed with PepMap C18, 2 μm particles, 100 Å pore size) using linear gradient from 4 to 35% acetonitrile containing 0.1% formic acid at a flow rate of 300 nl/min for 60 min.

Ionized peptides were analyzed on a Thermo Orbitrap Fusion (Q-OT-qIT; Thermo Scientific). Survey scans of peptide precursors from 350 to 1,400 $m/z$ were performed at 120K resolution settings with a $4 \times 10^5$ ion count target. Four different types of tandem MS were performed according to precursor intensity. First three types were detected in Ion trap in rapid mode, and last one was detected in Orbitrap with 15,000 resolution settings: (i) For precursors with intensity between $1 \times 10^3$ to $7 \times 10^3$ with CID fragmentation (35% collision energy) and 250 ms of ion injection time. (ii) For ions with intensity in range from $7 \times 10^3$ to $9 \times 10^4$ with CID fragmentation (35% collision energy) and 100 ms of ion injection time. (iii) For ions with intensity in range from $9 \times 10^4$ to $5 \times 10^6$ with HCD fragmentation (30% collision energy) and 100 ms of ion injection time. (iv) For intensities $5 \times 10^6$ and more with HCD fragmentation (30% collision energy) and 35 ms of ion injection time. The dynamic exclusion duration was set to 60 s with a 10 ppm tolerance around the selected precursor and its isotopes. Monoisotopic precursor selection was turned on. The instrument was run in top speed mode with 3 s cycles.

## Data analysis

All MS data were analyzed and quantified with the MaxQuant software (version 1.6.5.0) (Cox et al, 2014). The false discovery rate (FDR) was set to 1% for both proteins and peptides, and minimum length was specified as seven amino acids. The Andromeda search engine was used for the MS/MS spectra search against the murine Swiss-Prot database (downloaded from Uniprot on June 2019). Trypsin specificity was set as C-terminal to Arg and Lys, also allowing the cleavage at proline bonds and a maximum of two missed cleavages.

β-methylthiolation, N-terminal protein acetylation, carbamidomethylation, Met oxidation, and eventually Ser/Thr/Tyr phosphorylation, were included as variable modifications. Label-free quantification was performed using intensity-based absolute quantification (iBAQ) algorithm, which divides the sum of all precursor-peptide intensities by the number of theoretically observable peptides (Schwanhausser et al, 2011). Data analysis was performed using Perseus 1.5.2.4 software (Tyanova et al, 2016).

Complex stoichiometry was estimated based on iBAQ intensity ratio between individual components of the complex and IL-17RC. The PCAs and heat maps were constructed in programming language R. As source data, we used relative iBAQ values normalized to IL-17RC with addition of pseudocount 0.001. These values were log2-transformed prior to the analysis. The PCA was calculated using *prcomp* function. The heat map was constructed using *heatmap.2* function available in *gplots v3.0.1.1* package.

The analysis of ACT1 sequence to estimate the intrinsic disorder was performed using programs SPOT disorder (Hanson et al, 2017), MFDp2 (Mizianty et al, 2013), and IUPred2A (Meszaros et al, 2018).

## RNA sequencing experiments

For the RNA sequencing experiment, ST2 cells were grown in 6-well plates, washed, and stimulated with IL-17 (500 ng/ml) in the presence or absence of MRT67307 (2 μM) in DMEM supplemented with 0.5% FCS. Total RNA was extracted from the cells using RNeasy Mini Kit (Qiagen) with on column DNAse treatment exactly according the manufacturer's protocol. The quality and quantity of isolated RNA were evaluated using Nanodrop and TapeStation 2200 (Agilent Technologies). The ensuing RNA sequencing analysis was based on three independent experiments.

For each sample, 1 μg of total RNA (RIN > 7.0, rRNA ratio [28S:18S rRNA] > 1.0) was sent to Macrogen Inc. for library preparation and sequencing. Briefly, first TruSeq RNA stranded library was generated and then sequencing was performed on the Illumina NovaSeq6000 with 100 bp paired-ends configuration, with about 30M reads per sample. RNA was then quantified at the transcript-level using Salmon (0.13.1) (Patro et al, 2017). First, a reference mouse transcriptome from Gencode M21(GRCm38.p6) was created by combining the sequence of protein-coding genes and long non-coding RNAs. Then, the transcriptome was indexed using Salmon index with parameter "–gencode". Salmon quant was then run for each sample using parameters "–gcBias", "–validateMappings", and "–allowDovetail".

Transcript-level estimates from Salmon were summarized into gene level estimates using tximport (Soneson et al, 2015); all genes with at least 5 reads in at least 1 sample were retained for downstream analysis; with these parameters, 15,935 genes were obtained. Differential gene expression analysis was performed using DESeq2 (Love et al, 2014), using *independentFiltering* and using the Benjamini–Hochberg procedure to adjust $P$-values. Genes with adjusted $P$-value < 0.01 are reported as statistically significant.

## Quantitative real-time PCR

Cells were washed and incubated 30–60 min in DMEM supplemented with 0.5% FCS. In some experiments, cells were pretreated with TBK1/IKKε inhibitor MRT67307 (2 μM). Cells were stimulated with SF-IL-17 (500 ng/ml) for 2 h and RNA was extracted with TRIzol (Thermo Fisher Scientific) and purified using RNA clean & concentrator kit (Zymo Research). Reverse transcription was performed with RevertAid Reverse Transcriptase (Thermo Fisher Scientific) using Oligo(dT)$_{18}$ primers. Quantitative PCR was performed using SYBR Master Mix (Top-Bio in Figs 2F and 4F or Roche in Figs EV2D and EV3F) or and analyzed on LightCycler 480 (Roche). Data were normalized to GAPDH. Following primer pairs for were used:

mGapdh: 5′-TGCACCACCAACTGCTTAGC and 5′-GGCATGGACTGTGGTCATGAG
mCxcl1: 5′-CTTGAAGGTGTTGCCCTCAG and 5′-TGGGGACACCTTTTAGCATC
mCxcl2: 5′-CGGTCAAAAAGTTTGCCTTG and 5′-TCCAGGTCAGTTAGCCTTGC
mTnf: 5′-CCACCACGCTCTTCTGTCTAC and 5′-AGGGTCTGGGCCATAGAACT

mIl6: 5′-ATGGATGCTACCAAACTGGAT and 5′-TGAAGGACTC TGGCTTTGTCT

## Statistics

The indicated statistical analyses were performed using Prism (GraphPad Software). The experimental data that were analyzed by unpaired two-tailed Student's *t*-test were tested for normality as follows: In Figs 1C, 2F, and EV3F data, normality was confirmed using Kolmogorov–Smirnov test. In Figs 4F and EV2D, normal distribution was assumed based on Fig 2F. In the case of Figs 3B and EV3B, in which certain proteins were not detected and thus did not follow normal distribution, two-tailed nonparametric Mann–Whitney statistical tests were used.

# Data availability

The datasets produced in this study are available in the following databases: The mass spectrometry data: PRIDE PXD019020 (http://www.ebi.ac.uk/pride/archive/projects/PXD019020). RNA-Seq data: Gene Expression Omnibus GSE150410 (https://www.ncbi.nlm.nih.gov/geo/query/acc.cgi?acc = GSE150410).

Expanded View for this article is available online.

## Acknowledgements

We thank Pavel Talacko from the Laboratory of Mass Spectrometry, Biocev, Charles University, Faculty of Science, where proteomic and mass spectrometric analysis had been done. We thank Ladislav Cupak for technical assistance. We thank Zdenek Cimburek and Matyas Sima for cell sorting. This project was supported by a Czech Science Foundation grant (17-27355Y), EMBO Installation grant (4420) and Charles University grant (PRIMUS/20/MED/003) awarded to P.D., ERC grant (FunDiT) and SNSF Promys grant (IZ11Z0_166538) awarded to O.S., the Czech Science Foundation grant (18-24070Y) awarded to M.H., the Wellcome Trust Seed Award in Science (207769/A/17/Z) awarded to G.S., the Institutional Development Plan of University of Ostrava and The Ministry of Education, Youth and Sports (IRP03_2018-2020 and projects no. SGS02/LF/2017-2018 and SGS01/LF/2018-2019) and Charles University grant (UNCE/MED/016). H.D., S.J., D.K., and T.S. are students at the Faculty of Science, Charles University in Prague. S.K. is a student at the Faculty of Science, University of Ostrava.

## Author contributions

PD conceived the study. HD, SJ, DK, TS, MP, AU, AD, KR, OS and PD planned, performed, and analyzed experiments. SK, MH, VF, and GS performed RNA-Seq experiment and analyzed data. KH contributed to design and analysis of MS measurements. PD and OS wrote the manuscript. All authors commented on the manuscript draft.

## Conflict of interest

The authors declare that they have no conflict of interest.

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
