## [Review Process File · The EMBO Journal]

Systematic analysis of IL-17 receptor signalosome reveals a robust regulatory feedback loop

Peter Draber, Helena Draberova, Sarka Janusova, Daniela Knizkova, Tereza Semberova, Michaela Pribikova, Andrea Ujevic, Karel Harant, Sofija Knapkova, Matous Hrdinka, Viola Fanfani, Giovanni Stracquadanio, Ales Drobek, Klara Ruppova, and Ondrej Stepanek

DOI: [10.15252/embj.2019104202](https://doi.org/10.15252/embj.2019104202)

Corresponding author(s): Peter Draber (peter.draber@lf1.cuni.cz), Ondrej Stepanek (ondrej.stepanek@img.cas.cz)

Review Timeline:

Submission Date:	5th Dec 19
Editorial Decision:	10th Jan 20
Revision Received:	18th May 20
Editorial Decision:	3rd Jun 20
Revision Received:	13th Jun 20
Accepted:	17th Jun 20

Editor: Karin Dumstrei

Transaction Report:

Dear Dr. Draber,

Thank you for submitting your manuscript to The EMBO Journal. Your study has now been seen by two referees and their comments are provided below.

As you can see from the comments both referees find the analysis interesting. However, they also indicate that further experiments are needed to support the key conclusions. There are several concerns raised regarding the AP-MS approach as well that should be addressed. Should you be able to address the concerns raised in full then I would like to invite you to submit a revised version. I should add that it is EMBO Journal policy to allow only a single major round of revision and that it is therefore important to resolve the concerns raised at this stage.

Thank you for the opportunity to consider your work for publication. I look forward to your revision.

with best wishes

Karin

Karin Dumstrei, PhD
Senior Editor
The EMBO Journal

When assembling figures, please refer to our figure preparation guideline in order to ensure proper formatting and readability in print as well as on screen:
<http://bit.ly/EMBOPressFigurePreparationGuideline>

- a point-by-point response to the referees' comments, with a detailed description of the changes made (as a word file).

- a word file of the manuscript text.

- individual production quality figure files (one file per figure)

- a complete author checklist, which you can download from our author guidelines (<https://www.embopress.org/page/journal/14602075/authorguide>).

- Expanded View files (replacing Supplementary Information)

Further information is available in our Guide For Authors:

The revision must be submitted online within 90 days; please click on the link below to submit the revision online before 9th Apr 2020.

Link Not Available

Referee #1:

upon binding of IL-17A to its receptors via mass spectrometry. Their data establish that IL-17 shows a preference for activation and recruitment of TBK1 and IKK ϵ kinases to its receptor complex over other signaling events. Preventing TBK1/IKK ϵ activity increased NF- κ B and MAPK signaling induced by IL-17 but exerted only limited effects on TNF- α -induced signaling. Mechanistically, NEMO and TRAF6 enable activation and recruitment of both TBK1 and IKK ϵ to the IL-17R complex. These kinases phosphorylated the adaptor ACT1 at multiple sites in the disordered mid-part of the protein to limit the amount of recruited TRAF6. Therefore TBK1 and IKK ϵ serve an essential function in IL-17 pathway, acting via TRAF6 and NEMO.

The overall findings of this study are interesting and logically presented and represent an advance over existing data. Although part of the proposed model for the TBK1/IKK ϵ -mediated negative-feedback loop in the IL-17 pathway was previously shown by Qu et al., 2012, this study brings more clarity to this mechanism. Their proposed function of NEMO in IL-17 signaling, beyond simply regulating IKK α/β , is intriguing. However, this manuscript needs more experiments and controls to support their claims, particularly related to how NEMO mediates TBK1/IKK ϵ recruitment and activation. They also need to reconcile their findings with previously-published reports on this pathway.

1. The authors claim that NEMO regulates TBK1/IKK ϵ independently of IKK α/β activation. However, to more convincingly support this finding, the phosphorylation of TBK1 and IKK ϵ should be assessed upon IKK α/β inhibition. If IKK α/β are not involved, how does phosphorylation of TBK1 and IKK ϵ occur in the IL-17 pathway?
2. The paper suggests that NEMO recruits TBK1 and IKK ϵ indirectly through the TANK and NAP1 adaptors. If so, then re-expression of WT NEMO, but not of a deletion mutant unable to bind TANK (Chariot A et al. 2002) or NAP1, should restore phosphorylation and recruitment of TBK1 and IKK ϵ to the receptor complex. This should be straightforward to do.
3. In order to investigate the role of TBK1/IKK ϵ kinases in shaping IL-17 responses, the authors employed MRT67307, a pharmacological inhibitor of both TBK1 and IKK ϵ . This inhibitor can also affect autophagy (through inhibition of ULK1). Authors need to use a more specific approach to investigate the role of TBK1/IKK ϵ kinases in the expression of IL-17 target transcript (for example, siRNA).
4. Authors need to perform a kinetic analysis of IL-17 target gene transcription after TBK1/IKK inhibition since inhibition of TBK1/IKK activity leads to an increase in signaling inhibitors, I κ B α and A20, mRNA at 2 hours following IL-17 stimulation (Figure 2B). Both molecules are known to dampen NF- κ B-mediated signaling. Could this explain why in other studies, where IL-17 target genes were assessed at later time point (4 and 8 hours), TBK1/IKK β were shown to positively regulate IL-17 signaling ?
5. The concentration of IL-17A is 5 - 10 fold higher than typically used in the literature, which needs to be explained. Moreover, the authors should perform a dose response testing high and physiological concentrations.
6. The ACT1 western blot in Figure 4D is confusing. TRAF6 E3 ligase activity was shown to be required for IL-17-induced ACT1 phosphorylation (Qu et al., 2012) which does not seem the case in this figure. ACT1 phosphorylation seems to be higher in TRAF6 KO cells alone (empty vector) or reconstituted with mutant TRAF6 in comparison to TRAF6 KO cells reconstituted with WT TRAF6. This needs to be explained.
7. How do the authors explain the enrichment of ACT1 when comparing the composition of IL-17RSC in NEMO KO cells vs WT cells (Figure 5A) ?
8. Total protein levels of TBK1 and IKK ϵ are missing from the majority of western blot figures. This is essential to include as changes in their protein expression could affect their phosphorylation.
9. Although authors provided a reasonable explanation for the controversial role of TBK1/IKK ϵ when comparing their results with Tanaka et al. 2019, it is not clear to the reader what could explain the discrepancy between the presented results and the studies of Bulek et al. 2011 and Herjan et al. 2018. This should be addressed more fully in the Discussion.
10. The absence of TRAF5 enrichment in IL-17 receptor complex upon IL-17 stimulation is unexpected and should be highlighted in the Discussion, since TRAF5 was shown to be implicated in IL-17-mediated posttranscriptional mechanisms (Sun et al. 2011).

11. Is the described mechanism functionally relevant in vivo? It was previously shown by Bulek et al., 2011 that IKK α is required for IL-17-mediated neutrophilia and pulmonary inflammation in vivo and since TBK1 KO mice are lethal, authors could rely on MRT administration. For example, mice could be subjected to IL-17-driven pulmonary inflammation by injecting rIL-17 in the presence or absence of MRT. At minimum, this should be discussed.

Minor points

1. Authors should discuss why just one of the previously identified Act1 serine (Qu et al., 2012: Ser 147, Ser 209, Ser222 and Bulek et al., 2011: Ser 311) phosphorylated by TBK1/IKK α was found in this present study (Ser 209).

2. Authors should discuss the study by Shi et al. 2011 showing that IL-17 stimulated degradation of Act1 via β -TrCP, which was phosphorylation-dependent. Indeed, the present study shows a decrease in β -TrCP enrichment in IL-17R complex upon TBK1/IKK β deficiency. What is the part of this mechanism in TBK1/IKK β mediating control of IL-17 signaling? Wouldn't this mechanism be sufficient to explain the consequences of TBK1/IKK deficiency in the IL-17 signaling?

3. When comparing the composition of IL-17R complex in WT and TBK1/IKK ϵ DKO cells, authors observed a decrease in TRAF2 enrichment (Figure 3A). This point should be discussed since TRAF2 is an important component of IL-17 mediated posttranscriptional mechanism. Furthermore, this result is in line with studies demonstrating the role of TBK1/IKK β in controlling mRNA stability.

Referee #2:

Bone marrow-derived stroma cells of mouse origin (ST2) were activated with homodimeric recombinant IL-17 molecules comprising 6xHis, 2xStrep tag, and a 1xFlag tag at their N-terminus. Upon lysis in dodecyl maltoside lysis buffer, IL-17RA-IL-17RC receptor complexes were enriched via two steps of affinity purification and subjected to LC-MS analysis. Control samples corresponded to ST2 cells in which recombinant tagged molecules were added post-lysis (see specific comments below). As expected on the basis of previous studies (reviewed in Li et al. 2019), IL-17RA, IL-17RC, ACT1, TRAF6, TRAF2, A20, ABIN1, TAX1BP1, NEMO, IKK α , IKK β , TBK1, IKK ϵ and Cullin1 were identified. In addition, the adaptors TANK and NAP1 were found to be part the IL-17R pathway. Intensity-based absolute quantification showed that TBK1 and IKK ϵ were among the most abundant components of the IL-17RA-IL-17RC complex. Consistent with that last finding, IL-17 treatment strongly induced the phosphorylation of TBK1 and IKK ϵ at level comparable to those achieved via TNF or IL-1. In contrast, NF- κ B and MAPK were only weakly phosphorylated. Interestingly, transcriptional analysis showed that pharmacological inhibition of TBK1 and IKK ϵ enhanced the IL-17-mediated upregulation of a wealth of target genes, suggesting these two kinases inhibit IL-17 transcriptional responses. Using cells rendered deficient in (1) TBK1, (2) IKK ϵ and (3) in both TBK1 and IKK ϵ , the activity of these two kinases was found required to strongly inhibit IL-17-triggered signals. In contrast ablation of both TBK1 and IKK ϵ had no enhancing effect on TNF-induced signals. Comparative AP-MS analysis of the IL-17RA-IL-17RC complex in WT cells and in cells deficient in both TBK1 and IKK ϵ revealed a 2-fold enrichment of TRAF6 and of the LUBAC complex which in turn enhanced binding of the adaptors TAB1/2/3 and of the associated kinase TAK1 and NEMO. Therefore, these data suggest that TBK1 and IKK ϵ are activated upon IL-17 stimulation and mediate inhibition of IL-17 signaling by limiting the recruitment of TRAF6 and

LUBAC and effector kinases TAK1 and IKK. AP-MS analysis showed that the presence of both ACT1 and TRAF6 was mandatory for the recruitment of TBK1 and IKK ϵ . Moreover, NEMO was found to recruit TBK1 and IKK ϵ via the K63-linkage formed by TRAF6. Comparison of the composition of the IL-17RA-IL-17RC complex in WT and NEMO KO cells via MS showed that NEMO deficiency changed the composition of IL-17RSC. As expected, IKK α and IKK β were absent in NEMO deficient cells. These cells also showed enhanced recruitment of TRAF6 and TAB/TAK1 complex. This suggests that NEMO-mediated recruitment of TBK1 and IKK ϵ induce the release of TRAF6 from the complex. ACT1 associates with IL-17R via its C-terminal SEFIR domain and interacts with TRAF6 via its first 15 N-terminal amino acids. Analysis of the structure-function relationships existing at the level of ACT1 and mutation of all the multiple phosphorylation sites in the unstructured mid-part of ACT1 substantially increased signaling responses to IL-17 and strongly enhanced recruitment of TRAF6 to IL-17RSC. Accordingly, TBK1 and IKK ϵ likely phosphorylate the disordered mid-part of the ACT1 protein to limit the amount of recruited TRAF6. By documenting a novel negative regulatory loop, this comprehensive study that combines quantitative AP-MS and Crispr-Cas9-induced mutations explains the low intensity signals that are triggered by the IL-17R in comparison with other proinflammatory receptors. As outlined below, several issues linked to the AP-MS methods used by the authors need to be addressed prior to publication to fix key quantitative parameters.

Specific comments

1/ The approach used by the authors to capture the IL-17RA-IL-17RC complexes relies on the use of tagged recombinant IL-17 molecules. Therefore, it differs from most of the published quantitative AP-MS approaches used to decipher the signaling complexes of receptors and adaptors (see for instance Hein et al. Cell 163 712). In those studies, one chain of the receptor is tagged and negative controls correspond to untagged, WT cells in the very same state of activation. Such last approach permits to use solid statistics tools permitting to define high-confidence interactors based for instance on false discovery rate. The strategy used in the present paper is interesting in that it permits to focus on liganded IL-17RA-IL-17RC complexes. However, it does not permit to perform appropriate control pull down experiment using WT cells. As a consequence, the negative controls experiments performed by the authors consist in lysing cells prior to adding tagged recombinant IL-17 molecules. The authors should thus fairly discuss potential caveats of their approach. First, the control cells are not in the same activation state since they were not stimulated for 15 min with IL-17. Second, as shown on Figure 1B, when added post-lysis the tagged recombinant IL-17 molecules only bind the IL17RA chain. Therefore, a blatant lack of symmetry exists between the control and the test samples. This may greatly impact on the definition/quantitation of bona fide interactors in that the present approach does not allow to properly determine the intensities of background interactions.

2/ Counting the number of identified peptides is not an accurate way to quantify protein interactions (proteins don't have the same size...). Accordingly, the authors should modify Figures 1B, and 4A/C and used on Normalized intensities or IBAQ values. Moreover, in Figures 4A 'WT 0 min' is misleading and replaced by SF-IL-17 post-lysis.

3/ In the heat maps shown in Figure 3A and 5A, the unit of the scale needs to be indicated (linear or logX).

4/ Using iBAQ values, have the authors attempted to calculate the fraction of IL-17-engaged IL-17RA-IL-17RC complexes after 15 minutes of activation?

5/ All comparative analyses using interaction stoichiometry have been normalized on the IL-17RC molecule. Is there a rationale to use IL-17RC instead of IL-17RA?

7/ Do the many cell lines that were generated via CRISPR/Cas9 originate from a single clone of were

polyclonal? In the case they originate from a single clone, have the authors tested several of them to avoid potential inter-clonal variation?

8/ Supplementary Table 1. The intensity ratio columns appear to be IL17R/Ctrl rather than Ctrl/IL17R.

Referee #1:

upon binding of IL-17A to its receptors via mass spectrometry. Their data establish that IL-17 shows a preference for activation and recruitment of TBK1 and IKKε kinases to its receptor complex over other signaling events. Preventing TBK1/IKKε activity increased NF-κB and MAPK signaling induced by IL-17 but exerted only limited effects on TNF-α-induced signaling. Mechanistically, NEMO and TRAF6 enable activation and recruitment of both TBK1 and IKKε to the IL-17R complex. These kinases phosphorylated the adaptor ACT1 at multiple sites in the disordered mid-part of the protein to limit the amount of recruited TRAF6. Therefore, TBK1 and IKKε serve an essential function in IL-17 pathway, acting via TRAF6 and NEMO.

The overall findings of this study are interesting and logically presented and represent an advance over existing data. Although part of the proposed model for the TBK1/IKKε-mediated negative-feedback loop in the IL-17 pathway was previously shown by Qu et al., 2012, this study brings more clarity to this mechanism. Their proposed function of NEMO in IL-17 signaling, beyond simply regulating IKKα/β, is intriguing. However, this manuscript needs more experiments and controls to support their claims, particularly related to how NEMO mediates TBK1/IKKε recruitment and activation. They also need to reconcile their findings with previously-published reports on this pathway.

We would like to thank for positive evaluation of our study and for valuable comments which undoubtedly helped us to improve the manuscript. We performed additional experiments confirming the critical role of NEMO in TBK1/IKKε recruitment to IL-17RSC (as described in our point-by-point response below). Importantly, inspired by the reviewer's suggestions, we focused on the recruitment of TRAF6 and TRAF2 to the IL-17RC by TBK1/IKKε.

TRAF6 is required for the activation of MAPK and NF-κB signaling pathways. In this manuscript we show that TRAF6 recruitment is strongly inhibited by TBK1/IKKε-mediated phosphorylation of ACT1 in both human and murine cell lines. In accord, ablation and inhibition of TBK1/IKKε promotes recruitment of TRAF6 and enhanced activation of MAPK and NF-κB pathways. In contrast, TRAF2 is not required for proximal signaling. Instead, it promotes stabilization of mRNA via regulation of mRNA binding proteins such as ARID5A, HuR, splicing factor SF2, and endoribonuclease Regnase-1 (Amatya et al, 2018; Herjan et al, 2018; Herjan et al, 2013; Somma et al, 2015; Sun et al, 2011). Ablation of TBK1/IKKε largely inhibited TRAF2 recruitment to IL-17RSC (Fig 3A and newly modified 3B). Our new data (newly added Fig 3F) document that chemical inhibition of TBK1/IKKε activity even promotes TRAF2 recruitment. These experiments established that TBK1/IKKε are enabling TRAF2 recruitment to IL-17RSC in a kinase-independent manner.

Stimulation of TRAF2-deficient cells with IL-17 induces substantially lower level of target cytokines mRNA as compared to wild type cells (Amatya et al., 2018). In accord, while inhibition of TBK1/IKKε activity markedly enhanced the abundance of target mRNA induced upon IL-17, cells deficient in TBK1/IKKε that cannot efficiently recruit TRAF2 to the IL-17RSC were unable to promote mRNA stabilization (newly added Fig EV3F). These data demonstrated that TBK1/IKKε kinases have a dual role: enzymatic activity of TBK1/IKKε leads to the inhibition of TRAF6 recruitment, while the kinase activity-independent recruitment of TRAF2 leads to enhanced mRNA stability. In this regard, TBK1/IKKε kinases are switching the IL-17 signaling from activation of signaling pathways to mRNA stabilization. These new experiments are in good accord with published literature: TBK1/IKKε function as inhibitors of downstream proximal signaling (Qu et al, 2012), while they potentiate mRNA stabilization (Bulek et al, 2011; Herjan et al., 2018; Tanaka et al, 2019).

We highly appreciate this reviewer's insight, as the revised version of the manuscript reconciles previous seemingly contradictory data to resolve the role of TBK1/IKKε in IL-17 signaling complex.

1. The authors claim that NEMO regulates TBK1/IKKε independently of IKKα/β activation. However, to more convincingly support this finding, the phosphorylation of TBK1 and IKKε should be assessed upon IKKα/IKKβ inhibition. If IKKα/β are not involved, how does phosphorylation of TBK1 and IKKε occur in the IL-17 pathway?

TBK1 and IKKε are highly homologous kinases that are held in auto-inhibited state prior to activation, which is triggered by phosphorylation of the Ser172 located in the kinase domain activation loop. This event can be mediated by either TBK1/IKKε themselves or by related kinases IKKα/β (Larabi *et al*, 2013; Ma *et al*, 2012). Inhibition of IKKα/β or upstream kinase TAK1 prior to IL-17 stimulation had no effect on TBK1/IKKε activation. Only combined inhibition of IKKα/β and TBK1/IKKε prevented phosphorylation of TBK1/IKKε on Ser172 (newly added Fig EV4H). These results confirm that IKKα/β kinases might contribute, but are not required for TBK1/IKKε activation upon IL-17 stimulation.

*2. The paper suggests that NEMO recruits TBK1 and IKKε indirectly through the TANK and NAP1 adaptors. If so, then re-expression of WT NEMO, but not of a deletion mutant unable to bind TANK (Chariot A *et al*. 2002) or NAP1, should restore phosphorylation and recruitment of TBK1 and IKKε to the receptor complex. This should be straightforward to do.*

The TANK and NAP1 binding domain is localized within coiled-coil domain of human NEMO located between amino acids (AA) 200-250, which correspond to AA 201-248 of murine NEMO (Chariot *et al*, 2002) (Lafont *et al*, 2018). We reconstituted NEMO-deficient cells with NEMO(WT), NEMO(Δ201-248) or empty vector. While NEMO(WT) rescued TBK1/IKKε activation, NEMO(Δ201-248) or empty vector did not (newly added Fig EV4G).

NEMO KO cells expressing NEMO(Δ201-248) or empty vector were impaired in their ability to recruit TBK1/IKKε to IL-17RSC which was accompanied by the absence of ACT1 phosphorylation and enhanced TRAF6 recruitment, leading to increased activation of JNK and p38 signaling pathways as opposed to NEMO(WT) cells (newly added Fig EV5C and EV5D). These data are in accord with the major role of TBK1/IKKε in inhibition of TRAF6 recruitment and document crucial role of NEMO in regulating the assembly of IL-17RSC complex.

3. In order to investigate the role of TBK1/IKKε kinases in shaping IL-17 responses, the authors employed MRT67307, a pharmacological inhibitor of both TBK1 and IKKε. This inhibitor can also affect autophagy (through inhibition of ULK1). Authors need to use a more specific approach to investigate the role of TBK1/IKKε kinases in the expression of IL-17 target transcript (for example, siRNA).

We would like to thank the reviewer for pointing us in this direction. We noted that while inhibition of TBK1/IKKε activity led to massive increase of transcription of target genes, deletion of TBK1/IKKε did not (newly added Fig EV3F). However, both inhibition and ablation of TBK1/IKKε activity led to marked increase in activation of MAPK and NF-κB signaling pathways. These data show two roles of TBK1/IKKε kinases. On one hand, they phosphorylate ACT1 leading to release of TRAF6, as demonstrated in this manuscript. On the other hand, they have adaptor role that promotes stabilization of mRNA.

As correctly pointed by this reviewer, ablation of TBK1/IKKε led to decreased TRAF2 recruitment (Fig 3A and newly modified 3B). TRAF2 was described to potentiate mRNA stability upon IL-17 stimulation and deficiency in TRAF2 leads to decreased IL-17 induced mRNA levels of several cytokines (Amatya *et al.*, 2018). In contrast, inhibition of TBK1/IKKε activity even increased recruitment of TRAF2 (newly added Fig 3F).

Altogether, our data show that TBK1/IKKε are major regulators of IL-17 signaling. They inhibit TRAF6 recruitment to IL-17RSC and activation of MAPK and NF-κB by phosphorylating ACT1 and, at the same time, enable TRAF2 recruitment and mRNA stabilization in an enzymatic activity-independent manner. These results now neatly explain the discrepancy between present manuscript and previous report demonstrating that TBK1/IKKε are inhibitors of MAPK and NF-κB pathways (Qu *et*

al., 2012), and manuscripts showing that these kinases are promoting stabilization of mRNA (Bulek *et al.*, 2011; Herjan *et al.*, 2018; Tanaka *et al.*, 2019). If it were not for this reviewer's suggestion, we would miss this important link connecting all published papers together.

4. Authors need to perform a kinetic analysis of IL-17 target gene transcription after TBK1/ IKK inhibition since inhibition of TBK1/IKKε activity leads to an increase in signaling inhibitors, IκBα and A20, mRNA at 2 hours following IL-17 stimulation (Fig 2B). Both molecules are known to dampen NF-κB-mediated signaling. Could this explain why in other studies, where IL-17 target genes were assessed at later time point (4 and 8 hours), TBK1/ IKKb were shown to positively regulate IL-17 signaling?

RT-PCR analysis confirmed that mRNA of selected cytokines remained highly elevated upon prolonged stimulation of cells for 4 and 8 hours with IL-17 in the presence of TBK1/IKKε inhibitor as compared to IL-17 stimulation alone (newly added Fig EV2D).

The genes encoding signaling inhibitors IκBα and A20 are downstream targets of NF-κB and the induction of these negative regulators is directly proportional to the strength of signaling. For example, TNF is a very potent activator of signaling and cytokine production and at the same time it promotes strong induction of these inhibitors (Draber *et al.*, 2015).

5. The concentration of IL-17A is 5 - 10 fold higher than typically used in the literature, which needs to be explained. Moreover, the authors should perform a dose response testing high and physiological concentrations.

Our results showed that even a relatively high concentration of IL-17 cannot induce strong signaling response measured as activation of MAPK and NF-κB pathways, due to the potent negative feedback loop mediated by enzymatic activity of TBK1/IKKε. Our newly added data analyzing signaling response in WT versus TBK1/IKKε DKO cells show that this negative feedback loop is taking place over a range of IL-17 concentrations in both murine ST2 and human HeLa cell lines (newly added Fig EV2F and EV2H).

*6. The ACT1 western blot in Fig 4D is confusing. TRAF6 E3 ligase activity was shown to be required for IL-17-induced ACT1 phosphorylation (Qu *et al.*, 2012) which does not seem the case in this Fig. ACT1 phosphorylation seems to be higher in TRAF6 KO cells alone (empty vector) or reconstituted with mutant TRAF6 in comparison to TRAF6 KO cells reconstituted with WT TRAF6. This needs to be explained.*

We apologize for this confusing Figure. ACT1 is phosphorylated only in the presence of TRAF6(WT), but not enzymatically inactive TRAF6(C70A). The immunoprecipitation of IL-17RSC was carried using beads with anti-Flag antibody of mouse origin. ACT1 antibody is also of mouse origin. In the panel in Fig 4D, we used anti-mouse secondary antibody conjugated to horseradish peroxidase (HRP), causing a strong background signal, which might have confused the reviewer. We re-stained these membranes using TRUEblot secondary antibody (Rockland Immunochemicals) that detects only intact secondary antibody. The re-stained immunoblot is substantially clearer in showing that ACT1 phosphorylation in IL-17RSC requires TRAF6 ligase activity (newly modified Fig 4D). These data confirm the major role of TRAF6-mediated formation of nondegradative polyubiquitin linkages in recruiting NEMO and TBK1/IKKε kinases that subsequently phosphorylate Act1.

7. How do the authors explain the enrichment of ACT1 when comparing the composition of IL-17RSC in NEMO KO cells vs WT cells (Fig 5A)?

The enrichment of ACT1 was very minor (only 1.2 fold) as compared to TRAF6, which is enriched 8.4 times in NEMO KO as compared to WT cells. We mention the small change in ACT1 abundance in IL-17RSC isolated from NEMO KO versus NEMO WT cells in the result section.

8. Total protein levels of TBK1 and IKKε are missing from the majority of western blot Figs. This is essential to include as changes in their protein expression could affect their phosphorylation.

We re-stained the frozen membranes and samples from experiments in which the total TBK1 and IKKε staining was missing and added these panels to the corresponding Figures.

9. Although authors provided a reasonable explanation for the controversial role of TBK1/IKKε when comparing their results with Tanaka et al. 2019, it is not clear to the reader what could explain the discrepancy between the presented results and the studies of Bulek et al. 2011 and Herjan et.al 2018. This should be addressed more fully in the Discussion.

As discussed above, the newly added experiments show that TBK1/IKKε have a dual role in IL-17RSC assembly. On one hand, they phosphorylate ACT1 leading to the release TRAF6, while on the other they promote the recruitment of TRAF2 in enzymatic activity-independent manner (Fig 3). TRAF6 is a major activator of proximal signaling pathways and ablation or inhibition of TBK1/IKKε promotes IL-17 induced activation of MAPK and NF-κB signaling pathways. In contrast, knockout of TBK1/IKKε, but not inhibition of their activity, prevent TRAF2 recruitment, which is required for stabilization of mRNA of IL-17-induced proinflammatory cytokine (Amatya et al., 2018; Herjan et al., 2018; Herjan et al., 2013; Somma et al., 2015; Sun et al., 2011).

10. The absence of TRAF5 enrichment in IL-17 receptor complex upon IL-17 stimulation is unexpected and should be highlighted in the Discussion, since TRAF5 was shown to be implicated in IL-17-mediated posttranscriptional mechanisms (Sun et al. 2011).

In contrast to TRAF2 and TRAF6, we did not detect TRAF5 recruitment to IL-17RSC in any of our MS experiments. We cannot exclude that TRAF5 recruitment might be very weak and/or transient and below the detection limits of our method. We mentioned this issue in the discussion.

11. Is the described mechanism functionally relevant in vivo? It was previously shown by Bulek et al., 2011 that IKKε is required for IL-17-mediated neutrophilia and pulmonary inflammation in vivo and since TBK1 KO mice are lethal, authors could rely on MRT administration. For example, mice could be subjected to IL-17-driven pulmonary inflammation by injecting rIL-17 in the presence or absence of MRT. At minimum, this should be discussed.

We now mention in the discussion that our data are not validated by *in vivo* models yet. However, our results analyzing IL-17 signaling in both human and murine cell lines are clearly showing robust negative feedback loop mediated by the kinase activity of TBK1/IKKε. Therefore, our study argues against targeting TBK1/IKKε kinase activity to alleviate IL-17-mediated autoimmune diseases, as inhibition of these kinases unleashes the full potential of IL-17 in promoting transcription of inflammatory cytokines and their stabilization.

Minor points

1. Authors should discuss why just one of the previously identified ACT1 serine (Qu et al., 2012: Ser 147, Ser 209, Ser222 and Bulek et al., 2011: Ser 311) phosphorylated by TBK1/IKKi was found in this present study (Ser 209).

The likely explanation is in the different approaches used for analysis of ACT1 phosphorylation. Our study purified the signaling complex recruited to IL-17RSC. Bulek et al., 2011 directly immunoprecipitated ACT1 from IL-17 stimulated cells and subjected it to mass-spec analysis, while Qu et al., 2012 expressed recombinant fragments of ACT1 incubated with TBK1 or IKKε and identified whether TBK1/IKKε can phosphorylate these fragments via biochemical methods.

Interestingly, all the ACT1 phosphorylation sites identified to date are localized in the disordered region of ACT1 molecule between AA20-380, separating TRAF6 binding site and the SEFIR domains. In our model, the exact position of individual phosphorylation sites is not decisive for the signaling outcome. As ACT1 is phosphorylated on a number of residues, the combined effect of ACT1 phosphorylation leads to the repulsion between individual ACT1 molecules present within the complex. This prevents TRAF6 binding sites from coalescing and providing high avidity TRAF6 docking site (Fig 7 in the current manuscript).

To further test this hypothesis, we reconstituted ACT1 knockout cells with ACT1(WT) or ACT1(Δ 20-380) mutant lacking the disordered region. The activation of MAPK and NF- κ B signaling pathways upon IL-17 stimulation was markedly enhanced in cells expressing ACT1(Δ 20-380) as compared to ACT1(WT) (newly added Fig 6F). These data further support the hypothesis that this region of ACT1 is enabling control of its association with TRAF6 and combined phosphorylation of distinct Ser/Thr residues within this region would likely have similar inhibitory effect.

2. Authors should discuss the study by Shi et al. 2011 showing that IL-17 stimulated degradation of ACT1 via β -TrCP, which was phosphorylation-dependent. Indeed, the present study shows a decrease in β -TrCP enrichment in IL-17R complex upon TBK1/IKK deficiency. What is the part of this mechanism in TBK1/IKKb mediating control of IL-17 signaling? Wouldn't this mechanism be sufficient to explain the consequences of TBK1/IKK deficiency in the IL-17 signaling?

We prepared cells deficient in Cullin1. In contrast to NEMO or TBK1/IKK ϵ KO, these cells did not exhibit enhanced phosphorylation of JNK and p38 upon IL-17 stimulation, even though Cullin1-mediated degradation of I κ B was inhibited (newly added Fig EV6C-D). Although β -TrCP1/2 and Cullin1 are likely required for regulation of ACT1 level upon prolonged stimulation (Shi et al, 2011), our experimental evidence does not support the major role of Culin-1 in regulation of proximal IL-17 signaling.

3. When comparing the composition of IL-17R complex in WT and TBK1/IKK ϵ DKO cells, authors observed a decrease in TRAF2 enrichment (Fig 3A). This point should be discussed since TRAF2 is an important component of IL-17 mediated posttranscriptional mechanism. Furthermore, this result is in line with studies demonstrating the role of TBK1/IKKb in controlling mRNA stability.

As discussed above, we highly appreciate reviewer's suggestion to focus on the TRAF2 recruitment. The role of TBK1/IKK ϵ in recruiting TRAF2 independently on their enzymatic activity is striking and explains the discrepancy between our study and the study that focused on mRNA stabilization mediated by these kinases (Bulek et al., 2011; Herjan et al., 2018).

Referee #2:

Bone marrow-derived stroma cells of mouse origin (ST2) were activated with homodimeric recombinant IL-17 molecules comprising 6xHis, 2xStrep tag, and a 1xFlag tag at their N-terminus. Upon lysis in dodecyl maltoside lysis buffer, IL-17RA-IL-17RC receptor complexes were enriched via two steps of affinity purification and subjected to LC-MS analysis. Control samples corresponded to ST2 cells in which recombinant tagged molecules were added post-lysis (see specific comments below). As expected on the basis of previous studies (reviewed in Li et al. 2019), IL-17RA, IL-17RC, ACT1, TRAF6, TRAF2, A20, ABIN1, TAX1BP1, NEMO, IKK α , IKK β , TBK1, IKK ϵ and Cullin1 were identified. In addition, the adaptors TANK and NAP1 were found to be part the IL-17R pathway. Intensity-based absolute quantification showed that TBK1 and IKK ϵ were among the most abundant components of the IL-17RA-IL-17RC complex. Consistent with that last finding, IL-17 treatment strongly induced the phosphorylation of TBK1 and IKK ϵ at level comparable to those achieved via TNF or IL-1. In contrast, NF- κ B and MAPK were only weakly phosphorylated. Interestingly, transcriptional analysis showed that pharmacological inhibition of TBK1 and IKK ϵ enhanced the IL-17-mediated upregulation of a wealth of target genes, suggesting these two kinases inhibit IL-17 transcriptional responses. Using cells rendered deficient in (1) TBK1, (2) IKK ϵ and (3) in both TBK1 and IKK ϵ , the activity of these two kinases was found required to strongly inhibit IL-17-triggered signals. In contrast ablation of both TBK1 and IKK ϵ had no enhancing effect on TNF-induced signals. Comparative AP-MS analysis of the IL-17RA-IL-17RC complex in WT cells and in cells deficient in both TBK1 and IKK ϵ revealed a 2-fold enrichment of TRAF6 and of the LUBAC complex which in turn enhanced binding of the adaptors TAB1/2/3 and of the associated kinase TAK1 and NEMO. Therefore, these data suggest that TBK1 and IKK ϵ are activated upon IL-17 stimulation and mediate inhibition of IL-17 signaling by limiting the recruitment of TRAF6 and LUBAC and effector kinases TAK1 and IKK. AP-MS analysis showed that the presence of both ACT1 and TRAF6 was mandatory for the recruitment of TBK1 and IKK ϵ . Moreover, NEMO was found to recruit TBK1 and IKK ϵ via the K63-linkage formed by TRAF6. Comparison of the composition of the IL-17RA-IL-17RC complex in WT and NEMO KO cells via MS showed that NEMO deficiency changed the composition of IL-17RSC. As expected, IKK α and IKK β were absent in NEMO deficient cells. These cells also showed enhanced recruitment of TRAF6 and TAB/TAK1 complex. This suggests that NEMO-mediated recruitment of TBK1 and IKK ϵ induce the release of TRAF6 from the complex. ACT1 associates with IL-17R via its C-terminal SEFIR domain and interacts with TRAF6 via its first 15 N-terminal amino acids. Analysis of the structure-function relationships existing at the level of ACT1 and mutation of all the multiple phosphorylation sites in the unstructured mid-part of ACT1 substantially increased signaling responses to IL-17 and strongly enhanced recruitment of TRAF6 to IL-17RSC. Accordingly, TBK1 and IKK ϵ likely phosphorylate the disordered mid-part of the ACT1 protein to limit the amount of recruited TRAF6. By documenting a novel negative regulatory loop, this comprehensive study that combines quantitative AP-MS and Crispr-Cas9-induced mutations explains the low intensity signals that are triggered by the IL-17R in comparison with other proinflammatory receptors. As outlined below, several issues linked to the AP-MS methods used by the authors need to be addressed prior to publication to fix key quantitative parameters.

We thank the reviewer for assessing the novelty of our study and we appreciate the valuable recommendations how to further improve the manuscript.

Specific comments

1/ The approach used by the authors to capture the IL-17RA-IL-17RC complexes relies on the use of tagged recombinant IL-17 molecules. Therefore, it differs from most of the published quantitative AP-MS approaches used to decipher the signaling complexes of receptors and adaptors (see for instance Hein et

al. Cell 163 712). In those studies, one chain of the receptor is tagged and negative controls correspond to untagged, WT cells in the very same state of activation. Such last approach permits to use solid statistics tools permitting to define high-confidence interactors based for instance on false discovery rate. The strategy used in the present paper is interesting in that it permits to focus on liganded IL-17RA-IL-17RC complexes. However, it does not permit to perform appropriate control pull down experiment using WT cells. As a consequence, the negative controls experiments performed by the authors consist in lysing cells prior to adding tagged recombinant IL-17 molecules. The authors should thus fairly discuss potential caveats of their approach. First, the control cells are not in the same activation state since they were not stimulated for 15 min with IL-17. Second, as shown on Fig 1B, when added post-lysis the tagged recombinant IL-17 molecules only bind the IL17RA chain. Therefore, a blatant lack of symmetry exists between the control and the test samples. This may greatly impact on the definition/quantitation of bona fide interactors in that the present approach does not allow to properly determine the intensities of background interactions.

We agree with the reviewer that our approach is innovative and differs from most traditional approaches employed to study signaling complexes. We are convinced that the present methodology for IL-17RSC isolation and analysis has several key advantages over the traditional approaches:

(i) As noted by the reviewer, our methodology allows isolation of ligand-bound receptors that are part of assembled signaling complex, but not receptors that are not occupied by the ligand. This enables us to obtain clearer data. More importantly, it also allows us to study how the ligand is orchestrating the assembly of the complex. Binding of murine L-17 ligand to its two receptors is sequential. IL-17 is first strongly binding IL-17RA and only subsequently IL-17RC is recruited to the complex. IL-17 does not bind IL-17RC receptor in the absence of IL-17RA (Kuestner *et al*, 2007). In accord, adding IL-17 post-lysis did not lead to immunoprecipitation of IL-17RC or any other downstream signaling molecules, which provide evidence that all these molecules are specifically recruited to the complex only upon activation. Importantly, our post-lysis control shows that none of the identified components of the complex are just nonspecifically binding the ligand, as we do not identify them in non-stimulated samples. Cells lacking a core adaptor, ACT1, cannot recruit the downstream interacting molecules upon IL-17 stimulation (Figure 4A), even though the IL-17RC is detected in these cells. Similarly, IL-17RSC isolated from TRAF6-deficient cells contains IL-17RA, IL-17RC and ACT1, but no additional components of the complex. These experiments provide evidence for the specificity of the identified interactors and also establish the hierarchy of the IL-17RSC assembly.

(ii) The expression of tagged proteins is indeed a widely used approach, but it has a number of limitations on its own, which we can overcome using our innovative approach. First, adding a tag to a protein might modify its function, either by changing the conformation or by masking the interaction site with other components. Second, re-expression of the tagged proteins might lead to overexpression artefacts. Finally, isolation of tagged molecules might lead to the co-purification of the proteins that are not part of the signaling complex, but interact with bait proteins in completely different cellular compartment, such as in the endoplasmic reticulum or Golgi complex. Isolation of signaling complexes via tagged ligand that binds strongly to its plasma membrane-localized receptor avoids all these issues, as only signaling components actively recruited to the signaling complex are isolated.

(iii) The present system in which the cells are stimulated with tagged ligand can be easily used in a variety of cell lines without the need to introduce tagged construct.

The reviewer is correct by claiming that the present approach does not control for the activation state of the cell. Thus, we cannot formally exclude the possibility in this assay that the relatively weak IL-17 signaling induces changes in post-translational modification of some protein(s) which will somehow lead to their unspecific pull down. However, it is important to stress that the control used in this study, i.e. addition of tagged ligand post-lysis, allows to control for unspecific binding of background contaminants to the beads used for immunoprecipitation, protein tags or the unique epitopes at the

boundary between the protein and its tag and for interactions which occur only post-lysis. At the same time, we ensure endogenous expression of all the signaling proteins.

Overall, our approach and the traditional methods have both their theoretical advantages and drawbacks that should be carefully considered based on the particular scientific question. We believe our experimental approach is very appropriate to study the early events mediating the assembly of membrane-bound IL-17 RSC. Importantly, our conclusions based on mass-spec analysis of IL-17RSC were subsequently confirmed in functional studies. We modified the manuscript in order to highlight the advantages and potential caveats of the present system.

2/ Counting the number of identified peptides is not an accurate way to quantify protein interactions (proteins don't have the same size...). Accordingly, the authors should modify Figs 1B, and 4A/C and used on Normalized intensities or IBAQ values. Moreover, in Figs 4A 'WT 0 min' is misleading and replaced by SF-IL-17 post-lysis.

In the revised manuscript the Figures 1B, 4A and 4C indicate the iBAQ values to provide quantitative information about the abundance of the proteins in the sample and the number of unique peptides to provide qualitative information about the detected proteins. We corrected the label in Fig 4A.

3/ In the heat maps shown in Fig 3A and 5A, the unit of the scale needs to be indicated (linear or logX).

We added the scale to the Fig. 3A and Fig. 5A. It is a row-normalized Z-score calculated from \log_2 iBAQ values. The complete data processing for the construction of the heat maps are described in the Methods section.

4/ Using iBAQ values, have the authors attempted to calculate the fraction of IL-17-engaged IL-17RA-IL-17RC complexes after 15 minutes of activation?

The exact calculation of the fraction of engaged receptors is not possible in our system, as we do not know the quantity of membrane bound receptors (as opposed to the percentage of receptors localized intracellularly) prior to stimulation.

5/ All comparative analyses using interaction stoichiometry have been normalized on the IL-17RC molecule. Is there a rationale to use IL-17RC instead of IL-17RA?

The rationale for the normalization to IL-17RC is based on the biology of murine IL-17RSC assembly. First, dimeric IL-17 binds to one molecule of IL-17RA. Only subsequently, one molecule of IL-17RC is recruited to the complex, which leads to the recruitment of intracellular signaling proteins. IL-17RC alone cannot interact with IL-17 that is not associated with IL-17RA (Kuestner *et al.*, 2007). In accord, we noted that in our post-lysis samples, IL-17 binds only IL-17RA, but not IL-17RC. Whereas immunoprecipitated IL-17RA might originate from the ligand-engaged IL-17RA/IL-17RC receptor as well as from pre-signaling IL-17RA-IL-17 pulled down, IL-17RC uniquely represents the assembled IL-17RSC complexes in the 1:2:1 stoichiometry between IL-17RA:IL-17:IL-17RC (Ely *et al.*, 2009; Goepfert *et al.*, 2017; Liu *et al.*, 2013). For this reason, IL-17RC represents the ideal candidate for the normalization. We added the explanation for our choice in the results section.

7/ Do the many cell lines that were generated via CRISPR/Cas9 originate from a single clone or were polyclonal? In the case they originate from a single clone, have the authors tested several of them to avoid potential inter-clonal variation?

The panels shown in the manuscripts are based on knockout cell lines derived from one clone. Experiments showing the markedly changed signaling output in NEMO-deficient and TBK1/IKK ϵ -deficient cells were confirmed using two independent cell lines prepared using different target sgRNA. For the sake of clarity, we did not include these data in the manuscript. Instead, we validated our results using

two different cell lines, ST2 and HeLa, throughout the manuscript. The knockout in these cells were prepared using different sgRNA target sequences, which excludes off-target effects and documents that our findings are relevant in both murine and human cells. Importantly, the results obtained using knockout cell lines were complemented by reconstitution studies, demonstrating that the phenotype observed in knockout cells is due to the absence of studied protein.

However, thanks to this reviewer comment, we realized that the reconstitution experiment was missing for HOIP-deficient cell line. We therefore revised the manuscript to include also experiment in which HOIP-deficient cells are reconstituted with HOIP(WT) or empty vector. As expected, re-expression of HOIP markedly enhanced NF- κ B signaling upon IL-17 stimulation in these cells (newly added Figure EV3E).

8/ Supplementary Table 1. The intensity ratio columns appear to be IL17R/Ctrl rather than Ctrl/IL17R.

We thank the reviewer for spotting this mistake, which is corrected in the revised version of the manuscript.

References cited in the response to reviewers

- Amatya N, Childs EE, Cruz JA, Aggor FEY, Garg AV, Berman AJ, Gudjonsson JE, Atasoy U, Gaffen SL (2018) IL-17 integrates multiple self-reinforcing, feed-forward mechanisms through the RNA binding protein Arid5a. *Sci Signal* 11
- Bulek K, Liu C, Swaidani S, Wang L, Page RC, Gulen MF, Herjan T, Abbadi A, Qian W, Sun D *et al* (2011) The inducible kinase IKKi is required for IL-17-dependent signaling associated with neutrophilia and pulmonary inflammation. *Nat Immunol* 12: 844-852
- Chariot A, Leonardi A, Muller J, Bonif M, Brown K, Siebenlist U (2002) Association of the adaptor TANK with the I kappa B kinase (IKK) regulator NEMO connects IKK complexes with IKK epsilon and TBK1 kinases. *J Biol Chem* 277: 37029-37036
- Draber P, Kupka S, Reichert M, Draberova H, Lafont E, de Miguel D, Spilgies L, Surinova S, Taraborrelli L, Hartwig T *et al* (2015) LUBAC-Recruited CYLD and A20 Regulate Gene Activation and Cell Death by Exerting Opposing Effects on Linear Ubiquitin in Signaling Complexes. *Cell Rep* 13: 2258-2272
- Ely LK, Fischer S, Garcia KC (2009) Structural basis of receptor sharing by interleukin 17 cytokines. *Nat Immunol* 10: 1245-1251
- Goepfert A, Lehmann S, Wirth E, Rondeau JM (2017) The human IL-17A/F heterodimer: a two-faced cytokine with unique receptor recognition properties. *Sci Rep* 7: 8906
- Herjan T, Hong L, Bubenik J, Bulek K, Qian W, Liu C, Li X, Chen X, Yang H, Ouyang S *et al* (2018) IL-17-receptor-associated adaptor Act1 directly stabilizes mRNAs to mediate IL-17 inflammatory signaling. *Nat Immunol* 19: 354-365
- Herjan T, Yao P, Qian W, Li X, Liu C, Bulek K, Sun D, Yang WP, Zhu J, He A *et al* (2013) HuR is required for IL-17-induced Act1-mediated CXCL1 and CXCL5 mRNA stabilization. *J Immunol* 191: 640-649
- Kuestner RE, Taft DW, Haran A, Brandt CS, Brender T, Lum K, Harder B, Okada S, Ostrander CD, Kreindler JL *et al* (2007) Identification of the IL-17 receptor related molecule IL-17RC as the receptor for IL-17F. *J Immunol* 179: 5462-5473
- Lafont E, Draber P, Rieser E, Reichert M, Kupka S, de Miguel D, Draberova H, von Massenhausen A, Bhamra A, Henderson S *et al* (2018) TBK1 and IKKepsilon prevent TNF-induced cell death by RIPK1 phosphorylation. *Nat Cell Biol* 20: 1389-1399
- Larabi A, Devos JM, Ng SL, Nanao MH, Round A, Maniatis T, Panne D (2013) Crystal structure and mechanism of activation of TANK-binding kinase 1. *Cell Rep* 3: 734-746
- Liu S, Song X, Chrnyk BA, Shanker S, Hoth LR, Marr ES, Griffor MC (2013) Crystal structures of interleukin 17A and its complex with IL-17 receptor A. *Nat Commun* 4: 1888
- Ma X, Helgason E, Phung QT, Quan CL, Iyer RS, Lee MW, Bowman KK, Starovasnik MA, Dueber EC (2012) Molecular basis of Tank-binding kinase 1 activation by transautophosphorylation. *Proc Natl Acad Sci U S A* 109: 9378-9383
- Qu F, Gao H, Zhu S, Shi P, Zhang Y, Liu Y, Jallal B, Yao Y, Shi Y, Qian Y (2012) TRAF6-dependent Act1 phosphorylation by the I kappa B kinase-related kinases suppresses interleukin-17-induced NF-kappaB activation. *Mol Cell Biol* 32: 3925-3937
- Shi P, Zhu S, Lin Y, Liu Y, Liu Y, Chen Z, Shi Y, Qian Y (2011) Persistent stimulation with interleukin-17 desensitizes cells through SCFbeta-TrCP-mediated degradation of Act1. *Sci Signal* 4: ra73
- Somma D, Mastrovito P, Grieco M, Lavorgna A, Pignalosa A, Formisano L, Salzano AM, Scaloni A, Pacifico F, Siebenlist U *et al* (2015) CIKS/DDX3X interaction controls the stability of the Zc3h12a mRNA induced by IL-17. *J Immunol* 194: 3286-3294
- Sun D, Novotny M, Bulek K, Liu C, Li X, Hamilton T (2011) Treatment with IL-17 prolongs the half-life of chemokine CXCL1 mRNA via the adaptor TRAF5 and the splicing-regulatory factor SF2 (ASF). *Nat Immunol* 12: 853-860

Tanaka H, Arima Y, Kamimura D, Tanaka Y, Takahashi N, Uehata T, Maeda K, Satoh T, Murakami M, Akira S (2019) Phosphorylation-dependent Regnase-1 release from endoplasmic reticulum is critical in IL-17 response. *J Exp Med* 216: 1431-1449

Dear Peter,

Thanks for submitting your revised manuscript to The EMBO Journal. Your study has now been seen by the two referees.

The referees appreciate the introduced changes and support publication here. I am therefore pleased to let you know that we will accept the manuscript for publication here. Before doing so there are just a few editorial points that need to be addressed.

- You have at the moment 6 EV figures, but you can only have 5. Can you combine two of them maybe. If not, you can always add the figure to the appendix. See also guide to authors
- thanks for adding source data - will you please merge the files so that there is one file/folder per figure
- Please check Figure 4G the p-TBK1 lane. I think it is slightly misaligned in the figure.
- The call out to 4H-I is missing the number '4' - see page 10
- I have asked our publisher to do their pre-publication checks on the paper. They will send me the file within the next few days. Please wait to upload the revised version until you have received their comments.

That should be all! You can submit the revised manuscript using the link below. Please upload a point-by-point response as well

Let me know if you have any further questions

With best wishes

Karin

Karin Dumstrei, PhD
Senior Editor
The EMBO Journal

Further information is available in our Guide For Authors:

The revision must be submitted online within 90 days; please click on the link below to submit the revision online before 1st Sep 2020.

Referee #1:

In this revised manuscript, the authors have addressed my comments. Their study examines the assembly of the early signaling complex formed upon binding of IL-17A to its receptors. As stated earlier, their data established that IL-17 shows a preference for strong activation and recruitment of TBK1 and IKK ϵ kinases to its receptor complex over other signaling events. In this revised manuscript, authors bring more clarity to the mechanism by which NEMO activates and recruits TBK1/IKK ϵ to IL-17R. In addition, their new data revealed a more nuanced role of TBK1/IKK ϵ kinases in regulating IL-17 signaling. On one hand, these kinases inhibit TRAF6 recruitment to IL-17R and subsequent activation of MAPK and NF- κ B signaling. On the other hand, TBK1/IKK ϵ enable TRAF2 recruitment in an enzymatic activity-independent mechanism, yet remains to be resolved. Thus, this manuscript brings more clarity to the mechanism by which TBK1/IKK ϵ regulates IL-17 signaling.

Referee #2:

The authors have appropriately addressed the issues I raised as well as those raised by the second reviewer. In my view the manuscript can be accepted in its present form.

Dear Peter,

Thank you for submitting your revised manuscript to The EMBO Journal. I have now had a chance to take a look at the introduced changes and I am pleased to accept the manuscript for publication here.

One last thing - can you please check that size of the synopsis image is OK. It should be 550 pixels wide by [200-400} high. You can send it to us via email.

Congratulations on a nice study!

Best Karin

Karin Dumstrei, PhD
Senior Editor
The EMBO Journal

Please note that it is EMBO Journal policy for the transcript of the editorial process (containing referee reports and your response letter) to be published as an online supplement to each paper. If you do NOT want this, you will need to inform the Editorial Office via email immediately. More information is available here: http://emboj.embopress.org/about#Transparent_Process

Your manuscript will be processed for publication in the journal by EMBO Press. Manuscripts in the PDF and electronic editions of The EMBO Journal will be copy edited, and you will be provided with page proofs prior to publication. Please note that supplementary information is not included in the proofs.

Should you be planning a Press Release on your article, please get in contact with embojournal@wiley.com as early as possible, in order to coordinate publication and release dates.

If you have any questions, please do not hesitate to call or email the Editorial Office. Thank you for your contribution to The EMBO Journal.

** Click here to be directed to your login page: <http://emboj.msubmit.net>

Corresponding Author Name: Peter Draber

Manuscript Number: EMBOJ-2019-104202